# FHDM-KGE: Fuzzy Hierarchical Modeling and Dual Mixture-of-Experts for Knowledge Graph Embedding

## Abstract

Real world knowledge graphs (KGs) exhibit rich hierarchical structures, and effectively modeling such structures is crucial for learning high-quality representations and boosting downstream reasoning performance. However, existing hierarchy-aware KGE methods suffer from two key limitations: (i) hard layer assignment inevitably causes information loss for boundary or multi-role entities, and (ii) the neglect of relational cross-layer differences restricts the expressiveness of relation embeddings. To overcome these issues, we propose FHDM-KGE, a Fuzzy Hierarchical Modeling with Dual Mixture-of-Experts framework for knowledge graph embedding (KGE). First, we introduce a differentiable SpringRank-based fuzzy hierarchy that assigns entities to multiple layers with soft memberships, preserving multi-level semantics. Then, we design a dual MoE architecture: an entity-side MoE (EMoE) module gated by fuzzy memberships to capture intra-layer nuances, and a relation-side MoE (RMoE) module guided by head–tail hierarchical differences to model cross-layer relational patterns. The resulting entity and relation embeddings are scored with a ConvE decoder. Experiments on multiple public benchmarks demonstrate that FHDM-KGE consistently outperforms strong baselines, validating the effectiveness of combining fuzzy hierarchical modeling with dual MoE specialization.

## 1 Introduction

Knowledge Graphs (KGs) serve as structured repositories for real-world knowledge, precisely describing entities and their semantic connections in the form of *(head entity, relation, tail entity)* triplets Hogan et al. (2021). They provide artificial intelligence systems with rich and computable prior information. Reasoning techniques based on knowledge graphs have demonstrated exceptional value in numerous fields, including recommendation systems Jiang et al. (2024c), intelligent Q&A Chen et al. (2025), search engines Wang et al. (2024), and financial risk control. Knowledge graph embedding (KGE) techniques, as an essential component for reasoning with knowledge graphs, have been extensively studied in both academia and industry. Numerous high-quality KGE models have been proposed for various downstream tasks.

In fact, real-world knowledge is not flatly distributed in the graph structure Jiang et al. (2024b), but naturally exhibits hierarchical characteristics: from abstract concepts to concrete instances to localized components or attributes, entities are often at different semantic heights. For example, in the Generalized Encyclopedic Knowledge Atlas (GENKA), $animal$ is located under $organism$, and $mammal$ is a subcategory of $animal$. Fully exploiting and utilizing such hierarchical information not only enhances the semantic consistency of the representation, but also strengthens the inference of subclasses through the knowledge of the parent class in data sparse scenarios, which significantly improves the effect of link prediction, type summarization, and other tasks Chen et al. (2021). Current KGE methods can already capture hierarchical information in KG to some extent and further embed it to improve the performance of downstream tasks. However, the existing hierarchical embedding methods face two major limitations when applied: 1) **information loss caused by hard layering** and 2) **expression limitations due to the neglect of relational cross-layer differences**.

**Expression limitations caused by ignoring relational cross-layer differences.** Hierarchical information is not only present in the entities themselves but also in the hierarchical span of different relations connecting them Zhang et al. (2020b). Relation representations can also contain hierarchical information. For example, on the right side of Figure 1, the relations marked in different colors represent different hierarchical spans. For instance, in the triple *(Bat, Similar to, Mouse)*, *Similar to* is an intra-hierarchical relation, while in *(Bat, Belongs to, Mammal)*, *Belongs to* is an inter-hierarchical relation. Existing methods do not consider the different hierarchical levels of relations, which may lead to confusion between intra-hierarchical and inter-hierarchical logic, thereby reducing the precision of the representations.

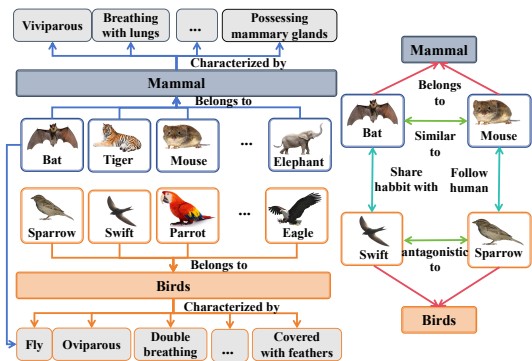

Figure 1: Examples illustrating the limitations of current KGE methods.

To address the two key issues mentioned above, we propose a novel method called **F**uzzy **H**ierarchical Modeling and **D**ual **M**ixture-of-Experts for **K**nowledge **G**raph **E**mbeddings (FHDM-KGE). The method can capture and represent the hierarchical information of entities and relations through differentiable fuzzy hierarchical structures and a dual Mixture-of-Experts (MoE) architecture. First, we use differentiable springrank to provide continuous hierarchical scores for each entity and assign entities to multiple levels with soft membership via fuzzy mapping, rather than rigidly to a single level, thereby capturing the possibility of entities spanning different levels. Then, we leverage existing GNN encoder frameworks to aggregate neighbor information and obtain basic entity and relation embeddings. Furthermore, we introduce two parallel MoE modules. The entity-side MoE (EMoE) module activates different hierarchical experts using fuzzy membership for entity representation embedding, while the relation-side MoE (RMoE) module determines the MoE for specific relations based on the hierarchical differences between head and tail nodes, thereby obtaining hierarchical embeddings for relations. Through this combination of mechanisms, FHDM-KGE ultimately generates entity and relation embeddings with better hierarchical information representation capabilities, thereby achieving improved performance in downstream tasks. Overall, the innovative contributions of this study include:

- We introduce a fuzzy soft hierarchical (**FH**) modeling mechanism based on the differentiable Springrank method. This mechanism can assign entities to multiple hierarchical levels with soft membership, thereby effectively capturing entities with multiple roles and alleviating the information loss caused by hard hierarchical division, which is conducive to learning better hierarchical information.

- We design two complementary expert modules: layer-specific entity MoE (**EMoE**) guided by fuzzy memberships, and relation MoE (**RMoE**) driven by head–tail hierarchical differences, enabling adaptive and specialized modeling of both entities and relations across layers. This allows for the simultaneous learning of entity and relation representations that contain hierarchical information, thereby improving the performance of downstream tasks.

- We formulate an integrated loss function that combines the standard KGE objective with hierarchy-consistency constraints and expert-balancing regularization, ensuring the model can be trained in a fully end-to-end manner. In addition, we have comprehensively validated our overall method through comparative experiments, ablation studies, sensitivity analyses, and case studies, demonstrating its superior performance.

## 2 RELATED WORKS

### 2.1 TRADITIONAL KGE METHODS

Traditional KGE approaches can be broadly categorized into translation-based models, semantic matching models, and graph neural network (GNN)-based models. Translation-based models (e.g.,

TransE Bordes et al. (2013), TransH Wang et al. (2014), TransR Lin et al. (2015)) represent a triple by enforcing the translation principle in a low-dimensional vector space. These models are computationally efficient and achieve good performance on simple relational patterns, but they struggle with complex many-to-many relations and cannot naturally encode asymmetric or hierarchical semantics. Semantic matching models (e.g., DistMult Yang et al. (2015), ComplEx Trouillon et al. (2016), HolE Nickel et al. (2016)) score triples by computing a similarity function, such as a bilinear product or complex-valued interaction—between entity and relation embeddings. These models offer greater flexibility in capturing symmetry, antisymmetry, and composition patterns, but they generally treat the KG as a flat structure and ignore global ordering constraints like hierarchy. GNN-based models (e.g., RGCN Schlichtkrull et al. (2018), CompGCN Vashishth et al. (2020), RGAT Busbridge et al. (2019)) incorporate multi-relational message passing, allowing entities to aggregate features from their neighbors through relation-specific transformations. Such methods effectively capture local structural context and multi-hop dependencies, yet they still lack explicit mechanisms to preserve hierarchical order, and the learned representations may conflate entities from different semantic levels. Overall, while these traditional KGE methods have advanced good performance in link prediction, their lack of explicit hierarchical modeling limits their ability to reason over KG with strong taxonomic or ontological structures.

## 2.2 HIERARCHY-AWARE KGE METHODS

To enhance downstream tasks, many methods have explored hierarchical modeling in KGs. HAKE Zhang et al. (2020a) introduced polar coordinate decomposition but focused mainly on entities. AttH Chami et al. (2020) placed embeddings in hyperbolic space with adaptive curvature and transformations. MSHE Jiang et al. (2024a) integrated structural and multi-hop information via a multi-source network. 3DH-KGE Lu et al. (2025) combined 3D rotation/translation with hyperbolic geometry. DHKE Zhang et al. (2024) used modulus in complex space with relation-specific scaling/rotation. HAQE Liang et al. (2024) and HRQE Yang et al. (2022) extended to quaternion space for unified relation–hierarchy modeling. SHLDKE Wang et al. (2025) mapped entities to a hypersphere for parameter-efficient hierarchical constraints. Due to the page limit, a detailed introduction of the existing Hierarchy-aware methods can be found in Appendix B.

Overall, these methods model the hierarchical information in KG from various perspectives and have achieved good performance in downstream tasks. However, existing methods still face limitations such as rigid hierarchical division leading to information loss and the neglect of hierarchical information in relations.

## 3 METHODOLOGY

### 3.1 OVERVIEW

As shown in Figure 2, we propose the FHDM-KGE method, which combines fuzzy soft hierarchical division with a dual MoE module Zhang et al. (2025). First, we use an RGCN encoder to obtain the initial embeddings of entities and relations. Then, we employ the differentiable SprinkRank method to learn the hierarchical scores of each entity and calculate their fuzzy memberships, thereby achieving fuzzy soft hierarchical division of entities. Next, we introduce a dual MoE module: entity-side mixture of experts (EMoE) and relation-side mixture of experts (RMoE). EMoE and RMoE activate and weight the experts based on the entity memberships and the hierarchical differences of the head and tail entities of relations, respectively, to obtain entity/relation representations that contain hierarchical information. Finally, we use a ConvE-based Dettmers et al. (2018) encoder to score the triples.

### 3.2 ENCODER

#### 3.2.1 BASIC ENCODING BASED ON RGCN

First, we employ an existing basic graph neural network encoder (RGCN) to obtain the basic representations of entities and relations through the message passing mechanism and local information aggregation. It is worth mentioning that this paper initializes the representations of entities and

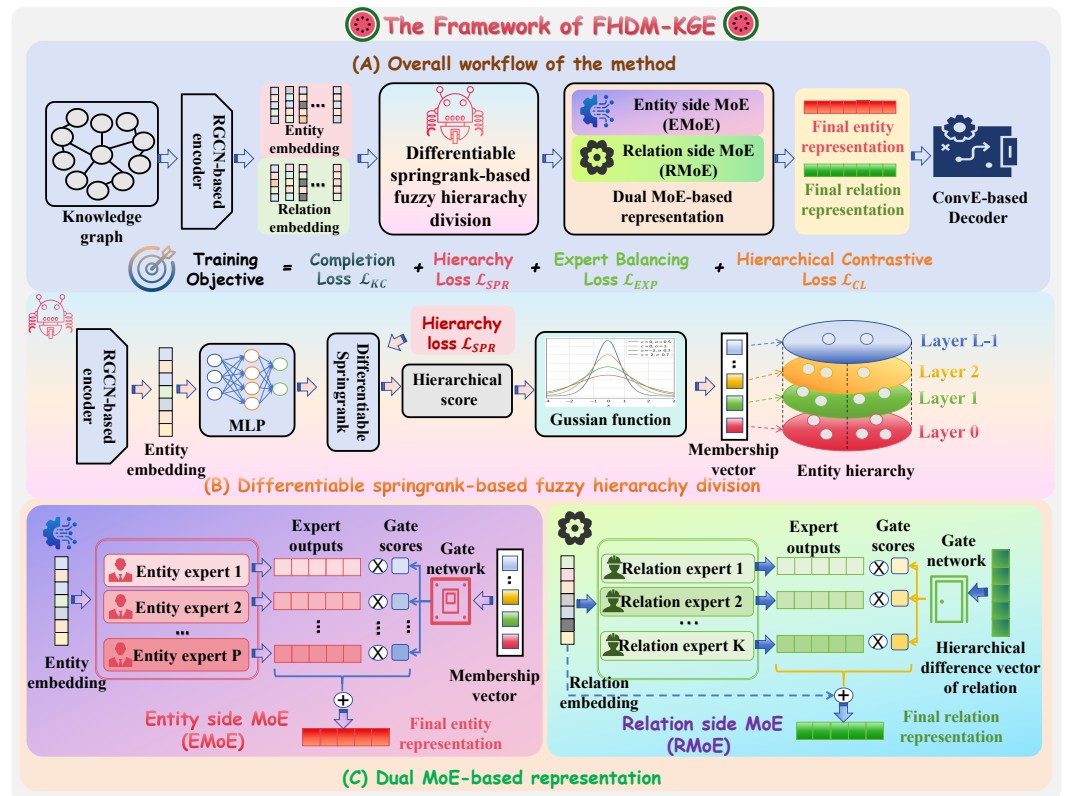

Figure 2: The framework of FHDM-KGE.

relations based on TransE, denoted as $\mathbf{e}^{(0)}$ and $\mathbf{r}^{(0)}$. The encoded entity representations is:

$$\mathbf{e}_i^{(t)} = \sigma\left(\mathbf{W}_0\,\mathbf{e}_i^{(0)} + \sum_{r\in\mathcal{R}} \sum_{j\in\mathcal{N}_r(e_i)} \frac{1}{|\mathcal{N}_r(e_i)|}\,\mathbf{W}_r^{(t-1)}\,\mathbf{e}_j^{(t-1)}\right), \tag{1}$$

where $\mathcal{N}_r(e_i)$ denotes the set of neighbors of entity $e_i$ connected by relation $r$. $\mathbf{W}_0 \in \mathbb{R}^{d^{(t)}\times d^{(0)}}$ is a trainable weight matrix for the self-loop (or entity's own features), and each $\mathbf{W}_r^{(t-1)} \in \mathbb{R}^{d^{(t)}\times d^{(t-1)}}$ is a trainable relation-specific weight matrix that transforms messages from a neighbor connected by relation $r$. We use $\sigma(\cdot)$ as an activation function, and $d^{(t)}$ is the output dimension of the RGCN layer.

### 3.2.2 Fuzzy Hierarchical Structure Based on Differentiable Springrank

Given RGCN-based encodings of entities and relations, $\mathbf{e}_i$ and $\mathbf{r}_z$, we obtain a continuous hierarchy score per entity via a small MLP: $s_i = \mathrm{MLP}(\mathbf{e}_i)$, where larger $s_i$ indicates a higher level. Unlike the closed-form SpringRank solution, these scores are learned end-to-end with the rest of the model. To regularize them, we adopt a SpringRank-inspired pairwise constraint: for each directed edge $e_u \to e_v$ we encourage $s_u \geq s_v + \delta$ (with a small margin $\delta \approx 1$). Aggregating over training triples $\mathcal{T}$, the hierarchy loss $\mathcal{L}_{\mathrm{SPR}}$ is defined as:

$$\mathcal{L}_{\mathrm{SPR}} = \sum_{(e_u\to e_v)\in\mathcal{T}} \log\left(1 + \exp\left(-(s_u - s_v - 1)\right)\right), \tag{2}$$

where the first term is a softplus (smooth) hinge that encourages $s_u - s_v \geq 1$ for every observed $e_u \to e_v$ edge (so that $e_u$ is ranked higher than $e_v$ by at least 1 unit). This is a differentiable approximation to the SpringRank objective. By minimizing $\mathcal{L}_{SPR}$, the model will learn $s$ values that reflect the directed structure of the KG: if a relation generally points from certain types of entities to others, the source entities' scores will be pushed higher than those of targets. Over many

triples, $s_i$ will tend to be larger for entities that often appear as heads of edges where the tails have lower scores, effectively learning a global ranking.

The continuous score $s_i$ for each entity is next transformed into a discrete but fuzzy layer membership. We decide on a fixed number of hierarchy layers $L$. Conceptually, $L$ could correspond to levels like "very specific" up to "very general", we define $L$ equally spaced target values between 0 and 1 to represent canonical layer positions:

$$\mu_l = \frac{l}{L-1}, \qquad l = 0, \ldots, L-1, \tag{3}$$

where $\mu_0 = 0$ corresponds to the bottom layer and $\mu_{L-1} = 1$ to the top layer, with intermediate $\mu_l$ evenly distributed. We then map each entity's raw score $s_i$ to a normalized layer membership vector $\mathbf{M}_i = (M_{i,0}, \ldots, M_{i,L-1})$. This vector is akin to a soft one-hot encoding over the $L$ layers, indicating the degree of belonging of entity $e_i$ to each layer:

$$M_{i,l} = \frac{\exp\left(-\frac{(\sigma(s_i)-\mu_l)^2}{2\sigma^2}\right)}{\sum_{q=1}^{L} \exp\left(-\frac{(\sigma(s_i)-\mu_q)^2}{2\sigma^2}\right)}, \quad l = 0, \ldots, L-1, \tag{4}$$

where $\sigma(\cdot)$ is sigmoid function. We treat each $\mu_l$ as the "center" of layer $p$ in the [0,1] interval. The membership $M_{i,l}$ is computed by a Gaussian kernel centered at $\mu_l$: it measures how close $z_i$ is to $\mu_l$, and then we normalize across all $p$ so that $\sum_l M_{i,l} = 1$ for each $e_i$. The bandwidth $\sigma^2$ in the Gaussian can be treated as a hyperparameter.

### 3.2.3 ENTITY-SIDE MIXTURE-OF-EXPERTS MODULE

After implementing the fuzzy hierarchical division of entities, in order to model hierarchical information in the final entity embedding representation, we further introduce a MoE mechanism on the entity side to learn different representations for entities of different hierarchies based on their membership degrees. First, we introduce $P$ entity side mixture of experts (EMoE) denoted as $\mathcal{W}_{e,1}, \mathcal{W}_{e,2}, \cdots, \mathcal{W}_{e,P}$, each of which uses a lightweight MLP network to obtain the corresponding expert output, as shown below:

$$\mathbf{e}_{i,p} = \mathcal{W}_p(\mathbf{e}_i) = \mathbf{W}_{p,2} ReLU(\mathbf{W}_{p,1}\mathbf{e}_i + b_{p,1}) + b_{p,2}. \tag{5}$$

Furthermore, we transform the membership degrees obtained from the hierarchical computation into soft gates to control the activation of different experts in the EMoE:

$$g_{i,p} = \frac{exp(\mathcal{Z}_e(\mathbf{e}_{i,p}, M_i) + \varepsilon_p)/\tau}{\sum_{j=1}^{P} exp(\mathcal{Z}_e(\mathbf{e}_{i,j}, M_i) + \varepsilon_j)/\tau}, \text{where} \quad \varepsilon_p \sim \mathcal{N}(0, \mathcal{Z}_e'(\mathbf{e}_{i,p}, M_i)), \tag{6}$$

where $\mathcal{Z}_e$ and $\mathcal{Z}_e'$ are two projection layers that map $(\mathbf{e}_{i,p}, M_i))$ to the mean and variance of the noisy gate, respectively, $\tau > 0$ is the temperature used to control the smoothness. Furthermore, we weight and aggregate the entity representations learned by each expert and the gate scores to obtain the final entity representation.

$$\mathbf{e}_i^f = \sum_{p=1}^{P} g_{i,p} \cdot \mathbf{e}_{i,p}, \tag{7}$$

where $g_{i,p}$ is the expert gate scores, $\mathbf{e}_i^f$ is the final entity representation obtained by aggregating the expert information based on the gating.

### 3.2.4 RELATION-SIDE MIXTURE-OF EXPERTS MODULE

After obtaining the entity embeddings containing hierarchical information through EMoE, we also introduce relation-side mixture of experts (RMoE). Unlike EMoE guided by membership degrees, we guide the relation experts by the hierarchical differences between the head and tail entities of the relation's triple. By introducing RMoE, we aim to enable the model to capture the hierarchical differences of relations, thereby enhancing the quality of relation embeddings.

First, we formally define the **layer difference** for a given triple $(h, r, t)$. Using the membership distributions for $h$ and $t$, we identify their most likely layers (or "peak" layers) as $\arg\max_l M_{h,l}$

and $\arg\max_l M_{t,l}$. Then: $\Delta(h,t) = \left| \arg\max_p M_{h,p} - \arg\max_p M_{t,p} \right|$, which yields an integer difference in layer indices. By definition $\Delta(h,t) \geq 0$. In practice, $\Delta(h,t)$ might range from 0 up to $L-1$. For each relation $r$, we compute a summary vector of its usage across these categories. Let $\mathcal{T}_r = \{(h,r,t) \in \mathcal{T}\}$ be the set of triples in the training set that involve relation $r$. We define a $L$-dimensional vector $\mathbf{G}(r) = [G_0(r), G_1(r), \cdots, G_{L-1}(r)]^\top$ where:

$$G_l(r) = \frac{1}{|\mathcal{T}_r|} \sum_{(h,r,t) \in \mathcal{T}_r} \mathbb{I}\big[\Delta(h,t) = l\big], \quad l \in \{0, 1, \cdots, L-1\}, \tag{8}$$

where $\mathbb{I}[\cdot]$ is the indicator function. $G_l(r)$ is basically the fraction of $r$'s triples that have layer difference $l$. For example, if relation $r$ usually connects same-layer entities, $G_0(r)$ will be high; if it often connects distant layers, $G_2(r)$ will be high, etc. $\mathbf{G}(r)$ can be viewed as a feature vector characterizing relation $r$ in terms of hierarchical jump pattern.

After statistically obtaining the hierarchical difference vectors of relations in the knowledge graph, similar to the entity expert learning in the previous section, we introduce $K$ experts as RMoE to learn the hierarchical information of relations, represented as: $\mathcal{W}_{r,1}, \mathcal{W}_{r,2}, \cdots, \mathcal{W}_{r,K}$. We also use an MLP to represent the transformation of the relation experts, the formula of which is:

$$\mathbf{r}_{i,k} = \mathcal{W}_k(\mathbf{r}_i) = \mathbf{W}_{k,2} ReLU(\mathbf{W}_{k,1}\mathbf{r}_i + b_{k,1}) + b_{k,2}, \tag{9}$$

where $\mathbf{r}_{i,k}$ represents the relation $r_i$ representation obtained after learning by the $k$ expert, $\mathbf{W}_{k,1}$ and $\mathbf{W}_{k,2}$ represent learnable matrixes, $\mathbf{r}_i$ represents the relation embeddings initialized by TransE. Furthermore, we construct a gating network based on the obtained hierarchical difference vectors of relations to obtain the weights for each relation expert as follows:

$$g_{i,k} = \frac{exp(\mathcal{Z}_r(\mathbf{r}_{i,k}, \mathbf{G}(r_i)) + \varepsilon_k)/\tau_r}{\sum_{o=1}^{K} exp(\mathcal{Z}_r(\mathbf{r}_{i,o}, \mathbf{G}(r_i)) + \varepsilon_o)/\tau_r}, \text{where} \quad \varepsilon_k \sim \mathcal{N}(0, \mathcal{Z}'_r(\mathbf{r}_{i,k}, \mathbf{G}(r_i))), \tag{10}$$

where $\mathcal{Z}_r$ and $\mathcal{Z}'_r$ are two projection layers, $\tau_r$ denotes the temperature, $g_{i,k}$ represents the weight of $k$ expert for $r_i$. Finally, we perform the weighted aggregation of the relation experts based on the obtained expert weights as shown below:

$$\mathbf{r}_i^f = \mathbf{W}_{ini}^r \mathbf{r}_i + \sum_{k=1}^{K} g_{i,k} \cdot \mathbf{r}_{i,k}, \tag{11}$$

where $\mathbf{r}_i^f$ represents the final embedding of relation $r_i$, $\mathbf{W}_{ini}^r$ is learnable matrix used to transform the initial embeddings to the same dimension as the final embeddings.

## 3.3 Decoder and Training Objective

After obtaining the final embeddings of entities and relations, we further introduce a decoder based on ConvE. For a given triple $\langle h, r, t \rangle$, its scoring function is defined as follows:

$$S(h,r,t) = ReLU(vec(ReLU((\overline{\mathbf{h}} \| \overline{\mathbf{r}}) * \omega)\mathbf{W}_c)\mathbf{t}, \tag{12}$$

where $\overline{\mathbf{h}}, \overline{\mathbf{r}} \in \mathbb{R}^{d_1 \times d_2}$ are the two-demensional reshaped vertors of $\mathbf{h}, \mathbf{r} \in \mathbb{R}^D$, where $D = d_1 \times d_2$ and $D$ is the dimension of entity and relation vectors. $\omega$ represents a set of filters, $*$ denotes the concolution operator, $vec(\cdot)$ is a vectorization function, $\mathbf{W}_c$ is the weight matrix.

In the link prediction task, the model aims to assign higher scores to positive triples and lower scores to negative triples. Therefore, we adopt the cross-entropy function as the loss for link prediction, as follows:

$$\mathcal{L}_{KGC} = \sum_{\langle h,r,t \rangle \in \Phi} -\frac{1}{|B|} \sum_{i=1}^{|B|} \chi_{h,r,t_i} \times log(S(h,r,t_i)) + (1 - \chi_{h,r,t_i}) \times log(1 - S(h,r,t_i)),$$
$$\tag{13}$$

where $\Phi$ is the set of positive triples, $|B|$ is the number of candidate entities, $S(h,r,t)$ is the score function obtained by ConvE, $\chi_{h,r,t_i}$ is the label of $\langle h, r, t_i \rangle$, $\chi_{h,r,t_i} = 0$ if the triple is negative and $\chi_{h,r,t_i} = 1$ if it is positive. To ensure the collaborative work of different components (fuzzy hierarchy, MoE) in the model, we introduce auxiliary loss terms on the basis of loss functions $\mathcal{L}_{KGC}$ and $\mathcal{L}_{SPR}$. These loss terms are used to maintain system consistency and prevent solution degeneration.

**Expert Usage Balancing Loss.**   We jointly balance (i) *layer usage* and (ii) *expert usage* on both entity- and relation-sides, while encouraging per-sample sparse routing. Let $\bar{\mathbf{m}} = \frac{1}{|\mathcal{B}|}\sum_{i\in\mathcal{B}}\mathbf{m}_i \in \Delta^{L-1}$, $\bar{\mathbf{g}}^E = \frac{1}{|\mathcal{B}|}\sum_{i\in\mathcal{B}}\mathbf{g}_i^E \in \Delta^{P-1}$, $\bar{\mathbf{g}}^R = \frac{1}{|\mathcal{B}|}\sum_{(h,r,\cdot)}\mathbf{g}^R \in \Delta^{K-1}$, and the uniform vectors $\mathbf{u}_L = \frac{1}{L}\mathbf{1}$, $\mathbf{u}_P = \frac{1}{P}\mathbf{1}$, $\mathbf{u}_K = \frac{1}{K}\mathbf{1}$. We minimize:

$$\mathcal{L}_{\text{EXP}} = \text{KL}(\bar{m} \,\|\, \mathbf{u}_L) \,+\, \text{KL}(\bar{g}^E \,\|\, \mathbf{u}_P) \,+\, \text{KL}(\bar{g}^R \,\|\, \mathbf{u}_K)\,. \tag{14}$$

The KL terms ensure global load-balancing; the entropy terms push individual routing to be low-entropy (i.e., sparse), complementing top-$k$ gating.

**Hierarchical Contrastive Loss.**   We perform a symmetric InfoNCE between $e_i$ and its layer mixture $p_i$, and add layer-aware prototype negatives. With in-batch negatives $\{p_j\}_{j\neq i}$ and prototype negatives $\{u_\ell\}_{\ell=1}^L$ weighted by $w_{i\ell} \overset{\triangle}{=} \beta\,(1 - m_{i,\ell})$ ($\beta \geq 0$), we define:

$$\mathcal{L}_{\text{CL}} = \frac{1}{2|B|}\sum_{i\in B}\left[ -\log\frac{\exp(\text{sim}(e_i, p_i)/\tau)}{\displaystyle\sum_{j\in B}\exp(\text{sim}(e_i, p_j)/\tau) \,+\, \sum_{\ell=1}^{L} w_{i\ell}\,\exp(\text{sim}(e_i, u_\ell)/\tau)} \right.$$
$$\left. -\log\frac{\exp(\text{sim}(p_i, e_i)/\tau)}{\displaystyle\sum_{j\in B}\exp(\text{sim}(p_i, e_j)/\tau)} \right], \tag{15}$$

where $\mathbf{p}_i = \sum_\ell m_{i,\ell}\mathbf{u}_\ell$; $\tau$ is the temperature; $\beta$ scales the penalty on off-layer prototypes via $w_{i\ell}$. Pull $\mathbf{e}_i$ toward its layer mixture while explicitly enlarging margins against other layers (via prototype-weighted negatives). Setting $\beta = 0$ reduces to standard in-batch contrastive learning.

**Overall Objective.**   We optimize:

$$\mathcal{L} = \mathcal{L}_{\text{KGC}} + \lambda_{\text{spr}}\mathcal{L}_{\text{SPR}} + \lambda_{\text{cl}}\mathcal{L}_{\text{CL}} + \lambda_{\text{exp}}\mathcal{L}_{\text{EXP}}, \tag{16}$$

where $\lambda_{\text{spr}}, \lambda_{\text{cl}}, \lambda_{\text{exp}}$ are hyperparameters controlling the relative weight of each auxiliary term.

## 4 EXPERIMENT

### 4.1 EXPERIMENT SETUP

**Dataset and Evaluation Protocol.**   We evaluated the proposed model on three commonly used knowledge graph datasets—FB15K-237 Toutanova et al. (2015), WN18RR Xiong et al. (2017), and YAGO3-10 Mahdisoltani et al. (2013). The detailed information of these datasets is summarized in Appendix C.2. We evaluate on the standard link prediction task: predicting the missing head or tail entity given a relation and the other entity. We use the filtered setting metrics: Mean Reciprocal Rank (MRR) and Hits@$K$ (for $K = 1, 3, 10$) of the correct entity in the ranked list of candidates.

**Baselines.**   To comprehensively evaluate the effectiveness of FHDM-KGE, we compared it with the following two categories of methods. Traditional embedding methods: DistMult Yang et al. (2015), ConvE Dettmers et al. (2018), ComplEx Trouillon et al. (2016), RotatE Sun et al., MGTCA Shang et al. (2024), and UniGE Liu et al. (2024); Hierarchy-Aware KGE Methods: HAKE Zhang et al. (2020a), MSHE Jiang et al. (2024a), ATTH Chami et al. (2020), 3DH-KGE Lu et al. (2025), DHKE Zhang et al. (2024), HAQE Liang et al. (2024), SHLDKE Wang et al. (2025).

### 4.2 MAIN RESULTS

On all three benchmark datasets, FHDM-KGE consistently outperforms both traditional and hierarchy-aware baselines across most evaluation metrics. In particular, our model achieves the highest MRR and Hits@K scores on FB15K-237, where it surpasses the strongest baseline (SHLDKE) by a large margin on Hits@1 (+5.3%) and also yields competitive improvements on Hits@3 and Hits@10. On the more challenging WN18RR dataset, FHDM-KGE establishes new state-of-the-art

Table 1: Link prediction results on FB15K-237, WN18RR and YAGO3-10 datasets, missing values are left blank, best results are in **bold**, and second best in underline.

| Models | FB15K-237 | | | | WN18RR | | | | YAGO3-10 | | | |
|---|---|---|---|---|---|---|---|---|---|---|---|---|
| | MRR | H@1 | H@3 | H@10 | MRR | H@1 | H@3 | H@10 | MRR | H@1 | H@3 | H@10 |
| DistMult | 0.241 | 0.155 | 0.263 | 0.419 | 0.430 | 0.390 | 0.440 | 0.490 | 0.340 | 0.240 | - | 0.540 |
| ConvE | 0.325 | 0.237 | 0.356 | 0.501 | 0.430 | 0.400 | 0.440 | 0.520 | 0.440 | 0.350 | 0.490 | 0.620 |
| ComplEx | 0.247 | 0.158 | 0.275 | 0.428 | 0.440 | 0.410 | 0.460 | 0.510 | 0.360 | 0.260 | 0.400 | 0.550 |
| RotatE | 0.338 | 0.241 | 0.375 | 0.533 | 0.476 | 0.428 | 0.492 | 0.571 | 0.495 | 0.402 | 0.550 | 0.670 |
| MGTCA | 0.393 | 0.291 | 0.401 | 0.583 | 0.511 | 0.475 | 0.525 | 0.594 | **0.586** | 0.514 | **0.629** | 0.721 |
| UniGE | 0.357 | 0.264 | 0.391 | 0.559 | 0.502 | 0.455 | 0.520 | 0.592 | 0.583 | 0.512 | 0.627 | 0.715 |
| HAKE | 0.346 | 0.250 | 0.381 | 0.542 | 0.497 | 0.452 | 0.516 | 0.582 | 0.545 | 0.462 | 0.596 | 0.694 |
| MSHE | 0.356 | 0.264 | 0.392 | 0.544 | 0.461 | 0.429 | 0.473 | 0.553 | 0.537 | 0.460 | 0.582 | 0.682 |
| ATTH | 0.324 | 0.236 | 0.354 | 0.501 | 0.466 | 0.419 | 0.484 | 0.551 | 0.397 | 0.310 | 0.437 | 0.566 |
| 3DH-KGE | 0.352 | 0.254 | 0.392 | 0.545 | 0.492 | 0.443 | 0.511 | 0.587 | - | - | - | - |
| DHKE | 0.356 | 0.260 | 0.392 | 0.548 | 0.494 | 0.453 | 0.509 | 0.576 | - | - | - | - |
| HAQE | 0.343 | 0.247 | 0.379 | 0.535 | 0.496 | 0.451 | 0.512 | 0.584 | 0.513 | 0.437 | 0.558 | 0.654 |
| SHLDKE | **0.398** | 0.278 | 0.402 | 0.556 | 0.502 | 0.487 | 0.515 | 0.586 | 0.566 | 0.443 | 0.612 | 0.712 |
| FHDM-KGE | 0.396 | **0.331** | **0.468** | **0.594** | **0.531** | **0.489** | **0.569** | **0.622** | 0.573 | **0.522** | 0.624 | **0.723** |

performance, obtaining an MRR of 0.531 and Hits@10 of 0.622, outperforming both flat models such as RotatE and hierarchy-enhanced models such as HAKE and DHKE. For YAGO3-10, our approach maintains comparable or superior results: while SHLDKE achieves the best Hits@10, FHDM-KGE delivers the best balance across MRR, Hits@1, and Hits@3, demonstrating strong robustness. Overall, these results confirm that integrating fuzzy hierarchical modeling with a dual mixture-of-experts design enables our model to capture complex hierarchical semantics and relation patterns more effectively than existing approaches.

## 4.3 ABLATION EXPERIMENTS

To validate the effectiveness of each module in FHDM-KGE, we conducted ablation studies from two dimensions: model design and loss function.

We performed KGC (Knowledge Graph Completion) experiments by removing the corresponding modules, and the results are shown in Table 2. Due to space limitations, we have placed the ablation experiment results and analysis on the other two datasets in Appendix C.3. "Full Model" represents our complete model. In the model design dimension, **w/o FH** means we removed the fuzzy hierarchy, which degenerates the model to hard hierarchy assignment; **w/o EMoE** means we removed the entity-side experts, replacing them with a single shared transformation; **w/o RMoE** means we removed the relation-side experts, using the base relation vector directly. In the loss function dimension,

Table 2: Ablation on FB15K-237. We separate *Model Design* and *Loss Design*. Columns follow the reference style: MRR / H@10 / H@3 / H@1. Best per column in **bold**.

| Setting | MRR | H@10 | H@3 | H@1 |
|---|---|---|---|---|
| w/o FH | 0.368 | 0.574 | 0.433 | 0.286 |
| w/o EMoE | 0.376 | 0.582 | 0.428 | 0.316 |
| w/o RMoE | 0.374 | 0.559 | 0.448 | 0.316 |
| w/o $\mathcal{L}_{SPR}$ | 0.376 | 0.584 | 0.450 | 0.309 |
| w/o $\mathcal{L}_{EXP}$ | 0.381 | 0.452 | 0.319 |
| w/o $\mathcal{L}_{CL}$ | 0.382 | 0.588 | 0.458 | 0.313 |
| **Full Model (FHDM-KGE)** | **0.396** | **0.594** | **0.468** | **0.331** |

**w/o $\mathcal{L}_{SPR}$** means we removed the hierarchical sorting constraints; **w/o $\mathcal{L}_{EXP}$** means we removed the expert balancing regularization; **w/o $\mathcal{L}_{CL}$** means we removed the hierarchical contrastive learning component.

**Fuzzy hierarchy is the primary source of gains.** Removing the fuzzy hierarchy leads to a clear drop. This confirms that hard assignments force *boundary / multi role* entities into a single layer, causing information loss; the effect is most visible on long-tail or abstract concepts.

**Dual experts are complementary.** Entity-side for within-layer refinement; relation-side for cross-layer adaptation. Removing *Entity-MoE* mainly hurts fine-grained discrimination within a layer, while removing *Relation-MoE* mainly weakens the modeling of cross-layer relations such as *typeOf/partOf/subsidiaryOf*.

**Loss design: every piece is necessary.** $\mathcal{L}_{SPR}$: Without it, the learned hierarchy score $s_i$ drifts and membership distributions become over sharp or over flat, hurting stability. $\mathcal{L}_{EXP}$: Without the expert-balancing regularizer, expert collapse emerges (lower routing entropy), causing small but

consistent drops. $\mathcal{L}_{CL}$: Removing the contrastive term makes same-level entities more confusable in prototype space, lowering Top-1 accuracy while tail metrics change less.

**Expert architecture analysis across datasets.** Beyond toggling the presence or absence of the entity-side and relation-side MoE modules, we further investigate whether the internal architecture of each expert is crucial for the gains of FHDM-KGE. We compare our full expert design—a two-layer non-linear transformation with hierarchy-aware conditioning and independent parameters for each expert—against three simplified variants on all three benchmarks: (1) *Linear experts*, where each expert is reduced to a single linear layer without non-linearity or bottleneck; (2) *w/o HierCond*, where experts no longer receive fuzzy-layer information and operate only on base embeddings; and (3) *Shared parameters*, where all experts share the same parameters and the gating network degenerates into a soft weighting of identical transformations. Due to space limitations, we have placed the experimental results and analysis in Appendix C.3.

### 4.4 HYPERPARAMETER SENSITIVITY ANALYSIS

The model is broadly robust within reasonable ranges, with each knob exhibiting a specific trade-off. (1) *Layers $L$*: too shallow underfits hierarchy; too deep adds noisy routing/overfitting; a moderate depth is best. (2) *Bandwidth $\sigma$*: very small values approach hard assignment and hurt multi-role entities; very large values blur layer separation; a mid-range preserves discrimination. (3) *Ranking weight $\lambda_{\text{spr}}$*: too weak fails to stabilize hierarchy; too strong over-regularizes and suppresses semantics; moderate weighting works best. (4) *Relation experts $M_r$*: improve cross-relation adaptation, but excessive experts cause routing instability and redundancy. (5) *Entity experts $M_e$*: sharpen intra-layer discrimination, with diminishing returns and higher cost when over-provisioned. (6) *Dimension $D$*: larger capacity helps up to a point, after which gains plateau and overfitting/ redundancy may appear. Overall, a moderate $L$, mid-range $\sigma$, balanced $\lambda_{\text{spr}}$, and compact $M_r, M_e, D$ yield the best accuracy–efficiency trade-off. Detailed FB15K-237 results appear in Appendix C.4.

### 4.5 CASE STUDY

To obtain a more intuitive understanding of how fuzzy hierarchy and dual MoE improve link prediction, we conduct a qualitative case study on FB15K-237. Figure 3 reports the Top-5 predictions of three hierarchy-aware models—our **FHDM-KGE**, **HAQE**, and **HAKE**—for four representative queries: (a) tail-entity prediction, (b) head-entity prediction, (c) cross-layer entity prediction, and (d) multi-hop prediction. For each query, we display both the ranked entities and their semantic types so that we can jointly evaluate the position of the gold answer and the quality of near-miss errors. Due to space limitations, we have placed the detailed analysis in Appendix C.5.

Across all four queries and three models, the qualitative evidence is consistent with our quantitative results: **FHDM-KGE not only ranks the gold entity higher in Top-5, but also produces semantically coherent near-miss candidates that stay within the correct type cluster**. **HAQE** yields intermediate behavior, while **HAKE** often ranks the gold lower and outputs off-type or wrong-hierarchy entities. This supports our claim that combining fuzzy hierarchical modeling with dual MoE leads to more accurate and more semantically disciplined link prediction.

### 4.6 EXPERT ROUTING AND SPECIALIZATION ANALYSIS

To go beyond aggregate link prediction metrics, we further analyze how the dual MoE modules behave on top of the learned fuzzy hierarchy. Concretely, we study (i) expert routing patterns across fuzzy layers for entities and across hierarchy spans for relations, and (ii) whether different experts specialize to distinct types of entities and relations as intended by our design. Due to space limitations, we have placed the detailed content in Appendix C.6.

### 4.7 COMPUTATIONAL EFFICIENCY ANALYSIS

In addition to the above experiments and analysis, to further enhance the persuasiveness of our method's performance, we conducted a computational efficiency analysis. We analyze the computational cost of our model from both a theoretical and an empirical perspective, and compare it with

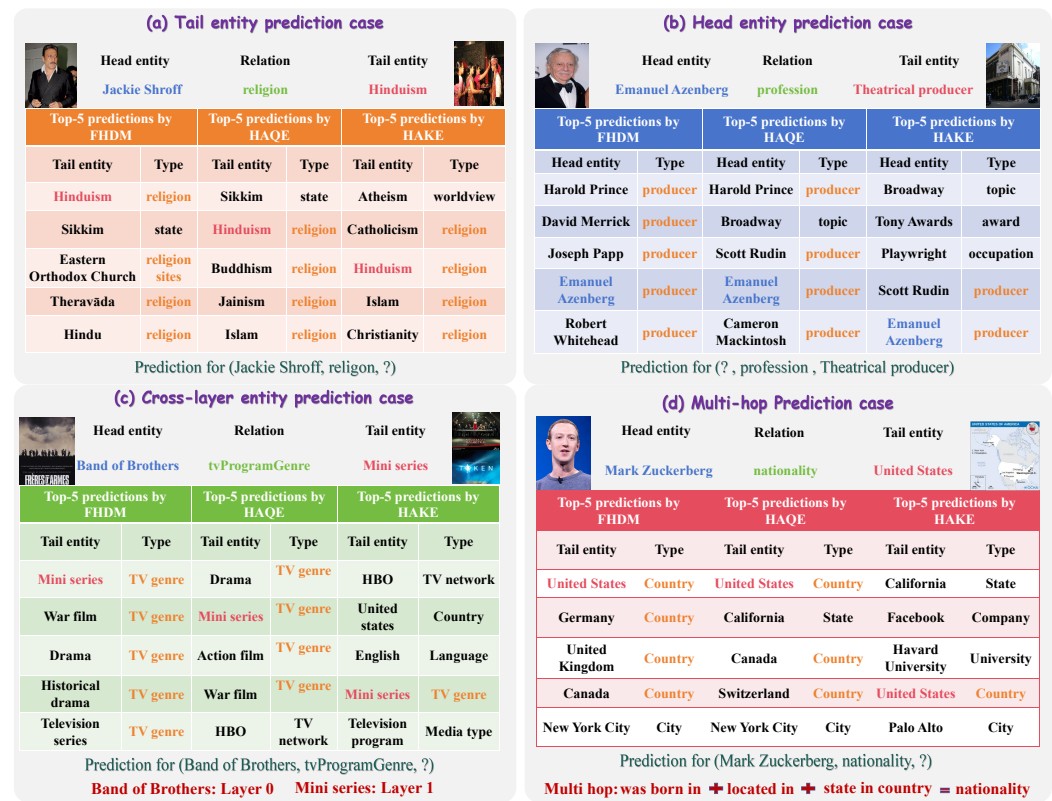

Figure 3: Case study of various queries on the FB15K-237 dataset using three methods (FHDM-KGE, HAQE, HAKE).

an RGCN+ConvE backbone under the same configuration (FB15K-237). Due to space limitations, we have placed the specific content in Appendix C.8.

## 4.8 OTHER STUDIES

In addition to the content mentioned above, we have elaborated in this section on some potential issues that may arise from the method we proposed. For example, the adaptability to imbalanced datasets, discussions on the joint gating of EMoE and RMoE, and theoretical analysis of the Springrank loss and fuzzy hierarchical interactions. Due to space limitations, the detailed content has been placed in Appendix C.9.

## 5 CONCLUSION

We proposed the FHDM-KGE knowledge graph embedding model, which combines differentiable hierarchical ranking with a layer-guided mixture-of-experts architecture. By endowing the model with the ability to infer and leverage latent hierarchical information, we addressed the key shortcomings of traditional knowledge graph modeling methods in handling hierarchical relationships. The design of FHDM-KGE enables it to adapt to different relational patterns: entities are represented using components matched to their level of abstraction, while relations are dynamically adjusted based on the span of hierarchy they cover. Extensive experiments have demonstrated that FHDM-KGE achieves state-of-the-art performance in link prediction tasks.

## 6 ETHICS STATEMENT

This work studies representation learning on publicly available knowledge graph benchmarks (e.g., FB15K-237, WN18RR, YAGO3-10). No human subjects were involved and no personally identifiable information (PII) or sensitive attributes are collected or generated beyond what is already contained in standard benchmarks. We adhere to the licenses accompanying these datasets and follow common community protocols (filtered ranking, link prediction splits).

## 7 REPRODUCIBILITY STATEMENT

We aim to make our results fully reproducible. All datasets used are standard and publicly accessible. The complete experimental protocol—including data preprocessing, train/validation/test splits, evaluation metrics (filtered ranking), early-stopping criteria, and the exact decoder/optimizer choices—is described in the paper and **Appendix**. **Importantly, we list all hyperparameters and their values used for each dataset in the Appendix,** together with search ranges and sensitivity analyses. We also specify random seed usage, batch sizes, number of epochs, and any task-specific settings. Upon acceptance, we will release: (i) the source code of our framework (training, evaluation, and ablation scripts), (ii) configuration files for all reported experiments, and (iii) instructions to reproduce the main tables and figures with a single command. These materials will enable exact replication of our reported numbers as well as straightforward extension to additional datasets and settings.

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

## A  THE USE OF LARGE LANGUAGE MODELS (LLMS)

We used large language models (LLMs), specifically OpenAI's ChatGPT, only for text editing and language refinement purposes. The models were employed to improve the clarity, readability, and fluency of the manuscript. All research ideas, methodology, experiments, analyses, and conclusions were developed entirely by the authors without assistance from LLMs. The LLMs did not contribute to generating novel content, designing experiments, or drawing scientific conclusions.

## B  RELATED WORKS

**Hierarchy-aware KGE Methods.**  In order to further enhance the performance of downstream tasks, many methods have begun to focus on modeling hierarchical information in KG. Early on, a representative method was HAKE Zhang et al. (2020a), which used polar coordinate decomposition in Euclidean space. The radius represented the hierarchical level, while the phase distinguished entities at the same level. However, it primarily focused on entities and lacked specific modeling for how relations behave differently across layers, often leading to insufficient expressiveness when dealing with multi-relational or multi-role entities. Subsequently, methods advanced along two main paths: geometry and capacity. AttH Chami et al. (2020) placed embeddings in hyperbolic space and used adaptive curvature for different relations, along with an attention mechanism to select geometric transformations like rotation or reflection. This approach was more hierarchy-friendly in low dimensions. However, the additional parameters for curvature and transformations increased complexity, and the curvature effect tended to weaken in higher dimensions. In parallel, MSHE Jiang et al. (2024a)integrated structural and multi-hop contextual information through a multi-source hierarchical network, significantly improving hierarchical discriminability and robustness. Yet, it had weaker geometric interpretability and higher training and tuning costs. To integrate hierarchy and complex relational patterns within a unified framework, 3DH-KGE Lu et al. (2025)used 3D rotation and translation combined with hyperbolic geometry to simultaneously express non-commutative relations and hierarchical structures. However, achieving stable rotation and translation training in hyperbolic geometry is highly complex from an engineering perspective. DHKE Zhang et al. (2024) further advanced this by using the modulus in complex space to represent hierarchy, with relation-specific scaling and rotation to modulate head and tail entities. This enabled the a priori learning of hierarchies and adaptation to different relations. However, splitting the complex vector dimensions and parameters introduced additional overhead and tuning burdens. To increase the degree of freedom, methods like HAQE Liang et al. (2024) and HRQE Yang et al. (2022) extended embeddings to quaternion space, using the modulus plus three-dimensional angles to unify the modeling of various relations and hierarchies. While this improved performance, it also significantly increased the number of parameters and the risk of overfitting, making them more dependent on regularization and search strategies. More recently, SHLDKE Wang et al. (2025) attempted to place entities on a unit hypersphere to compress dimensions and improve parameter efficiency. It leveraged positive curvature and bounded volume to reflect hierarchical constraints. However, its fit for deep tree-like structures and its expressive capacity remained limited, and it often involved trade-offs with other relational patterns.

## C  EXPERIMENT

### C.1  DETAILS OF DATASETS

Table 3: Statistics of datasets used in experiments. Train, Valid, and Test represent the number of training, validation, and test queries, respectively.

| Dataset | Entity | Relation | Train | Valid | Test |
|---|---|---|---|---|---|
| WN18RR | 40.9k | 11 | 21.7k | 3.0k | 3.1k |
| FB15k-237 | 14.5k | 237 | 68.0k | 17.5k | 20.4k |
| YAGO3-10 | 123.1k | 37 | 269.7k | 5.0k | 5.0k |

## C.2 Implementation Details

In our experiments, we implemented our model method using PyTorch and tested it on a Linux server running Ubuntu 24.04.2, equipped with two NVIDIA A6000 GPUs. During training, the batch size was selected from {1024, 2048}, and the embedding dimension was chosen from { 300, 400, 500}. The model was trained using the Adam optimizer, with the learning rate selected from {1e-3, 5e-4, 3e-4}. The experimental results for baseline methods were reproduced according to the settings in their original papers and their open-source code. More detailed implementation specifics are shown in the Table 4.

Table 4: Hyperparameter setting for diffrent datasets.

| Hyperparameters | FB15K-237 | WN18RR | YAGO3-10 |
|---|---|---|---|
| Batchsize | 1024 | 1024 | 2048 |
| Epoch | 2000 | 2000 | 2500 |
| Learning rate | 5e-4 | 3e-4 | 1e-3 |
| Layers $L$ | 4 | 4 | 4 |
| Bandwidth $\sigma$ | 0.20 | 0.25 | 0.25 |
| Relation Experts $M_r$ | 3 | 3 | 3 |
| Entity Experts $M_e$ | 4 | 4 | 4 |
| Dim $D$ | 300 | 500 | 400 |
| $\lambda_{\text{spr}}$ | 0.60 | 0.50 | 0.50 |
| $\lambda_{\text{cl}}$ | 1.0 | 1.0 | 1.0 |
| $\lambda_{\text{exp}}$ | 0.5 | 0.5 | 0.5 |
| Optimizer | Adam | Adam | Adam |

## C.3 Ablation Study

Table 5: Ablation on WN18RR and YAGO3-10. We separate *Model Design* and *Loss Design*. Columns follow the reference style: MRR / H@10 / H@3 / H@1. Best per column on each dataset in **bold**.

| Setting | WN18RR | | | | YAGO3-10 | | | |
|---|---|---|---|---|---|---|---|---|
| | MRR | H@10 | H@3 | H@1 | MRR | H@10 | H@3 | H@1 |
| w/o FH | 0.493 | 0.601 | 0.526 | 0.423 | 0.532 | 0.699 | 0.577 | 0.451 |
| w/o EMoE | 0.504 | 0.609 | 0.520 | 0.467 | 0.544 | 0.708 | 0.571 | 0.498 |
| w/o RMoE | 0.502 | 0.585 | 0.545 | 0.467 | 0.541 | 0.680 | 0.597 | 0.498 |
| w/o $\mathcal{L}_{\text{SPR}}$ | 0.504 | 0.612 | 0.547 | 0.456 | 0.544 | 0.711 | 0.600 | 0.487 |
| w/o $\mathcal{L}_{\text{EXP}}$ | 0.511 | 0.609 | 0.550 | 0.471 | 0.551 | 0.708 | 0.603 | 0.503 |
| w/o $\mathcal{L}_{\text{CL}}$ | 0.512 | 0.616 | 0.557 | 0.462 | 0.553 | 0.716 | 0.611 | 0.494 |
| **Full Model (FHDM-KGE)** | **0.531** | **0.622** | **0.569** | **0.489** | **0.573** | **0.723** | **0.624** | **0.522** |

From Table 5, we observe that the ablation trends on WN18RR and YAGO3-10 are highly consistent with those on FB15K-237. First, removing the fuzzy hierarchy (**w/o FH**) causes the largest degradation on both datasets: on WN18RR, MRR drops from 0.531 to 0.493 and Hits@1 from 0.489 to 0.423; on YAGO3-10, MRR decreases from 0.573 to 0.532 and Hits@1 from 0.522 to 0.451. This confirms that explicitly modeling fuzzy hierarchical structure is crucial for capturing multi-level semantics rather than being a dataset-specific trick. Second, eliminating either the entity-side experts (**w/o EMoE**) or the relation-side experts (**w/o RMoE**) also leads to clear performance drops. On WN18RR, both variants lose around 0.027–0.029 MRR and 0.022 Hits@1, while on YAGO3-10 they lose 0.029–0.032 MRR and 0.024 Hits@1. Notably, **w/o RMoE** yields a more pronounced degradation in Hits@10 (e.g., 0.723→0.680 on YAGO3-10), suggesting that relation-side experts are particularly important for ranking a large set of candidate tails, whereas entity-side experts contribute more to sharpening top-ranked predictions.

In terms of loss design, removing any of the three objectives consistently harms performance, but to different extents. The hierarchical sorting loss **w/o $\mathcal{L}_{\text{SPR}}$** produces the largest degradation among the loss ablations on both WN18RR (MRR 0.531→0.504, Hits@1 0.489→0.456) and YAGO3-10 (MRR 0.573→0.544, Hits@1 0.522→0.487), indicating that enforcing an ordered fuzzy hierarchy is essential for fully exploiting the learned layers. Removing the expert balancing regularizer (**w/o**

$\mathcal{L}_{\textbf{EXP}}$) and the hierarchical contrastive loss (**w/o** $\mathcal{L}_{\textbf{CL}}$) also leads to stable but slightly smaller drops, showing that both terms help avoid expert collapse and encourage hierarchy-aware discrimination. Overall, across all three benchmarks (FB15K-237, WN18RR, and YAGO3-10), the full FHDM-KGE model consistently achieves the best results, and each proposed component—fuzzy hierarchy, dual MoE, and the three losses—contributes non-trivially, demonstrating the robustness and generality of our design.

Table 6: Expert architecture ablation on FB15K-237, WN18RR, and YAGO3-10. Columns per dataset: MRR / H@10 / H@3 / H@1.

| Setting | FB15K-237 | | | | WN18RR | | | | YAGO3-10 | | | |
|---|---|---|---|---|---|---|---|---|---|---|---|---|
| | MRR | H@10 | H@3 | H@1 | MRR | H@10 | H@3 | H@1 | MRR | H@10 | H@3 | H@1 |
| Linear experts (1-layer, no HierCond) | 0.382 | 0.580 | 0.452 | 0.309 | 0.514 | 0.607 | 0.552 | 0.472 | 0.553 | 0.708 | 0.606 | 0.503 |
| w/o HierCond inside experts | 0.387 | 0.587 | 0.456 | 0.318 | 0.519 | 0.612 | 0.556 | 0.478 | 0.558 | 0.714 | 0.610 | 0.508 |
| Shared expert parameters | 0.389 | 0.585 | 0.457 | 0.320 | 0.522 | 0.614 | 0.555 | 0.480 | 0.562 | 0.713 | 0.611 | 0.511 |
| **Full experts** | **0.396** | **0.594** | **0.468** | **0.331** | **0.531** | **0.622** | **0.569** | **0.489** | **0.573** | **0.723** | **0.624** | **0.522** |

**Expert Architecture Analysis.** From Table 6, across all three datasets, the full expert architecture consistently achieves the best performance, while simplifying the experts leads to noticeable degradation. On FB15K-237, reducing experts to a single linear layer ("Linear experts") decreases MRR from 0.396 to 0.382 and Hits@1 from 0.331 to 0.309, indicating that non-linear depth is important for modeling complex hierarchical interactions. Removing hierarchy-aware conditioning inside experts ("w/o HierCond") or sharing parameters across experts ("Shared expert parameters") yields slightly better results than the linear variant but still lags behind the full model (e.g., MRR 0.387–0.389 and Hits@1 0.318–0.320), showing that both fuzzy-layer conditioning and expert diversity contribute to the overall gains.

A similar pattern holds on WN18RR and YAGO3-10. On WN18RR, MRR decreases from 0.531 to 0.514 when using linear experts and remains below the full model for all simplified variants, with Hits@1 dropping from 0.489 to 0.472. On YAGO3-10, the full experts reach 0.573 MRR and 0.522 Hits@1, while the best simplified variant (shared parameters) remains lower at 0.562 MRR and 0.511 Hits@1. Overall, the consistent gaps between the full and simplified variants across datasets indicate that the performance improvements of FHDM-KGE do not stem from the mere presence of an MoE layer, but from the specific internal design of the experts: non-linear transformations, hierarchy-aware conditioning, and independent expert parameters are all necessary to fully exploit the fuzzy hierarchical structure.

## C.4 HYPERPARAMETER SENSITIVITY ANALYSIS

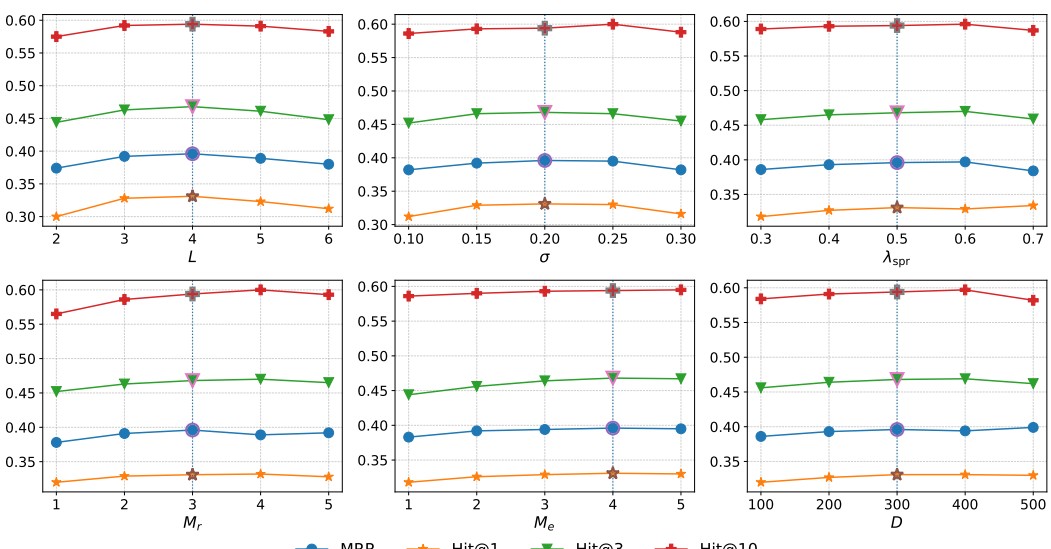

Figure 4: Hyperparameter sensitivity on FB15K-237

Table 7: Hyperparameter sensitivity on FB15K-237.

| Layers $L$ | | | | | Bandwidth $\sigma$ | | | | | Ranking weight $\lambda_{\text{spr}}$ | | | |
|---|---|---|---|---|---|---|---|---|---|---|---|---|---|
| Value | MRR | H@10 | H@3 | H@1 | Value | MRR | H@10 | H@3 | H@1 | Value | MRR | H@10 | H@3 | H@1 |
| 2 | 0.374 | 0.575 | 0.444 | 0.300 | 0.10 | 0.382 | 0.586 | 0.452 | 0.312 | 0.30 | 0.386 | 0.589 | 0.458 | 0.318 |
| 3 | 0.392 | 0.592 | 0.463 | 0.328 | 0.15 | 0.392 | 0.593 | 0.466 | 0.329 | 0.40 | 0.393 | 0.588 | 0.465 | 0.327 |
| **4** | **0.396** | **0.594** | **0.468** | **0.331** | **0.20** | **0.396** | 0.594 | **0.468** | **0.331** | **0.50** | 0.396 | **0.594** | 0.468 | **0.331** |
| 5 | 0.389 | 0.591 | 0.461 | 0.323 | 0.25 | 0.395 | **0.600** | 0.466 | 0.330 | 0.60 | **0.397** | 0.590 | **0.470** | 0.329 |
| 6 | 0.380 | 0.583 | 0.448 | 0.312 | 0.30 | 0.382 | 0.588 | 0.455 | 0.316 | 0.70 | 0.384 | 0.587 | 0.459 | 0.327 |

| Relation experts $M_r$ | | | | | Entity experts $M_e$ | | | | | Embedding dim $D$ | | | |
|---|---|---|---|---|---|---|---|---|---|---|---|---|---|
| Value | MRR | H@10 | H@3 | H@1 | Value | MRR | H@10 | H@3 | H@1 | Value | MRR | H@10 | H@3 | H@1 |
| 1 | 0.378 | 0.565 | 0.452 | 0.320 | 1 | 0.383 | 0.586 | 0.444 | 0.318 | 100 | 0.386 | 0.584 | 0.456 | 0.320 |
| 2 | 0.391 | 0.586 | 0.463 | 0.329 | 2 | 0.392 | 0.590 | 0.456 | 0.326 | 200 | 0.393 | 0.591 | 0.464 | 0.327 |
| **3** | **0.396** | 0.594 | **0.468** | **0.331** | 3 | 0.391 | 0.594 | 0.464 | 0.329 | 300 | **0.396** | 0.594 | **0.468** | **0.331** |
| 4 | 0.389 | **0.600** | 0.462 | 0.327 | **4** | **0.396** | 0.594 | **0.468** | **0.331** | 400 | 0.394 | **0.597** | **0.469** | 0.330 |
| 5 | 0.392 | 0.593 | 0.465 | 0.316 | 5 | 0.389 | **0.596** | 0.459 | 0.330 | 500 | 0.388 | 0.582 | 0.462 | 0.330 |

Across FB15K–237, we observe a clear "Goldilocks" pattern with respect to the number of fuzzy layers $L$, fuzziness bandwidth $\sigma$, SpringRank regularization weight $\lambda_{\text{spr}}$, the counts of entity- and relation-side experts $(M_e, M_r)$, and the embedding dimension $D$. As $L$ increases from 2 to 4, MRR rises steadily (e.g., from 0.374 to 0.396), but degrades at $L \in \{5, 6\}$, indicating that excessive depth introduces noise and overfitting; thus $L = 4$ is preferable. For $\sigma$, the range 0.20-0.25 yields the best or near-best MRR ( 0.395-0.396): smaller $\sigma$ degenerates toward hard partitioning, while larger $\sigma$ weakens inter-layer separability. The hierarchy-preserving constraint also benefits from moderation: $\lambda_{\text{spr}} \in [0.50, 0.60]$ maximizes overall performance—$\lambda_{\text{spr}} = 0.60$ gives the highest MRR (0.397), where $\lambda_{\text{spr}} = 0.50$ slightly improves Hits@10 (0.594), reflecting a trade-off between stable ranking and top-$k$ retrieval. On the mixture-of-experts design, relation-side MoE peaks at $M_r = 3$ (larger $M_r$ destabilizes routing and adds redundancy), while entity-side MoE attains its best at $M_e = 4$ (further increases deliver diminishing returns). Capacity-wise, $D = 300$ offers the strongest overall balance—larger dimensions (e.g., $D \in \{400, 500\}$) do not translate into robust gains and can slightly reduce MRR. In summary, a compact yet sufficient configuration is recommended: $L = 4$, $\sigma \in [0.20, 0.25]$, $\lambda_{\text{spr}} \in [0.50, 0.60]$, $M_r = 3$, $M_e = 4$, and $D = 300$. These trends corroborate our intuition that (i) moderate depth and fuzziness retain multi-role entity information without over-regularization; (ii) a mid-strength hierarchical ranking prior stabilizes layer structure without suppressing complementary semantics; and (iii) carefully bounded expert counts and dimensionality avoid routing collapse and parameter redundancy while preserving expressivity.

## C.5 CASE STUDY

**(a) Tail-entity prediction.** For the query (*Jackie Shroff, religion, ?*), the gold tail is *Hinduism*. **FHDM-KGE** ranks *Hinduism* at the top and keeps the remaining candidates within the religion family (e.g., *Sikhism, Theravāda, Eastern Orthodox Church*), producing near-miss errors that are semantically coherent with the gold type. **HAQE** also includes *Hinduism* and several other religions, but the gold is ranked slightly lower and the candidate set mixes in a state entity (e.g., *Sikkim*), reflecting weaker type consistency. **HAKE** performs worst: although it eventually includes *Hinduism*, it ranks it lower and its Top-5 contains more heterogeneous belief systems (e.g., *Atheism, Catholicism, Islam, Christianity*) and even a state, leading to a noisier prediction list.

**(b) Head-entity prediction.** For the reverse profession query (*?, profession, Theatrical producer*), all three models retrieve some correct theatrical producers. **FHDM-KGE** provides the cleanest candidate set: its Top-5 consists entirely of real theatrical producers (e.g., *Harold Prince, David Merrick, Joseph Papp, Emanuel Azenberg, Robert Whitehead*), and ranks the gold entity *Emanuel Azenberg* among these peers. **HAQE** places *Emanuel Azenberg* and several genuine producers in its Top-5 but also introduces off-type items such as the topic *Broadway*, again showing intermediate quality. **HAKE** is the noisiest: it ranks non-person entities such as *Broadway* and *Tony Awards* ahead of producers, and repeatedly surfaces off-type occupations (e.g., *Playwright*), indicating that it confounds the profession relation with loosely related cultural concepts.

**(c) Cross-layer entity prediction.** The query (*Band of Brothers, tvProgramGenre, ?*) links a concrete TV mini-series at the instance layer to an abstract genre at a higher layer. Our fuzzy hierarchy explicitly treats this as a cross-layer relation. **FHDM-KGE** correctly ranks *Mini series* at the top and fills the rest of the list with closely related TV/film genres (e.g., *War film, Drama, Historical drama,*

*Television series*), all of which share the same "genre/type" semantic level. **HAQE** also retrieves *Mini series* but usually at a lower rank and mixes in more generic genres (e.g., *Action film*) and a channel-type entity (*HBO*), partially blurring the boundary between "program genre" and "broadcast network". **HAKE** struggles most: its Top-5 is dominated by *HBO*, *United States*, *English*, and *Television program*, i.e., TV networks, countries, and media types rather than genres, and only occasionally ranks *Mini series* within Top-5, showing that it has difficulty separating instance-to-genre cross-layer semantics from other contextual neighbors.

**(d) Multi-hop prediction.** The query (*Mark Zuckerberg*, *nationality*, ?) requires multi-hop reasoning along paths such as *works at $\rightarrow$ located in state $\rightarrow$ state in country* to reach *United States*. **FHDM-KGE** again ranks *United States* at the top and keeps all remaining candidates within the same semantic type (other countries such as *Germany*, *United Kingdom*, *Canada*), demonstrating that the learned fuzzy hierarchy and relation-side experts successfully integrate multi-hop and cross-layer cues. **HAQE** achieves intermediate performance: it usually places *United States* near the top, but its Top-5 also contains a mixture of countries, states (e.g., *California*), and cities (e.g., *New York City*), indicating partial confusion between different geographic levels. **HAKE** performs the worst: it often promotes local neighbors such as *California*, *Facebook*, *Harvard University*, and *Palo Alto* ahead of the true country, and its error set is dominated by states, cities, and institutions rather than countries, revealing a strong bias toward shallow structural neighbors.

## C.6 EXPERT ROUTING AND SPECIALIZATION ANALYSIS

**Entity-side routing across fuzzy layers.** We first examine the entity-side MoE by aggregating gating weights over entities whose dominant fuzzy layer is $L_\ell$. For each dataset, we compute the layer-wise expert distribution:

$$P(p^* \mid L_\ell) = \frac{1}{|\mathcal{E}_\ell|} \sum_{e \in \mathcal{E}_\ell} g_e(p^* \mid e), \tag{17}$$

where $\mathcal{E}_\ell$ collects entities whose main membership lies in layer $L_\ell$, and $g_e(\cdot)$ denotes the entity-side gating network. The resulting heatmaps in Fig. 5 (top row) show a clear diagonal structure: each fuzzy layer $L_\ell$ tends to prefer one or two experts, and this pattern is consistent on FB15K-237, WN18RR, and YAGO3-10. In particular, lower layers (e.g., $L_0$ and $L_1$) exhibit sharply peaked routing onto a single expert, while higher layers (e.g., $L_2$ and $L_3$) mix multiple experts more heavily. To quantify this effect, we report the average gating entropy:

$$H_{\text{ent}}(L_\ell) = -\frac{1}{|\mathcal{E}_\ell|} \sum_{e \in \mathcal{E}_\ell} \sum_p g_e(p \mid e) \log g_e(p \mid e), \tag{18}$$

normalized by the maximum entropy $\log 4$ in Figure 6 (top row). Across all datasets, $H_{\text{ent}}(L_0)$ and $H_{\text{ent}}(L_1)$ are substantially lower than $H_{\text{ent}}(L_2)$ and $H_{\text{ent}}(L_3)$, confirming that entities near the bottom of the hierarchy are handled by more specialized experts, whereas high-level, more "global" entities benefit from distributing probability mass over multiple experts.

Analogously, we analyze the relation-side MoE as a function of the hierarchy span between head and tail. For each relation triple $(h, r, t)$, we approximate the span $\Delta$ by the distance between the dominant layers of $h$ and $t$, and bucket it into intra-layer ($\Delta{=}0$), adjacent-layer ($\Delta{=}1$) and long-range ($\Delta{\geq}2$) cases. We then compute:

$$P(k^* \mid \Delta) = \frac{1}{|\mathcal{T}_\Delta|} \sum_{(h,r,t) \in \mathcal{T}_\Delta} g_r(k^* \mid r, h, t), \tag{19}$$

where $\mathcal{T}_\Delta$ groups triples with span $\Delta$ and $g_r(\cdot)$ is the relation-side gating network. As shown in Figure. 5 (bottom row), short-range edges ($\Delta{=}0$) consistently activate one dominant expert, while adjacent and long-range edges ($\Delta{=}1$ and $\Delta{\geq}2$) gradually shift probability mass towards different experts, indicating that the relation-side MoE learns to specialize not only by relation semantics

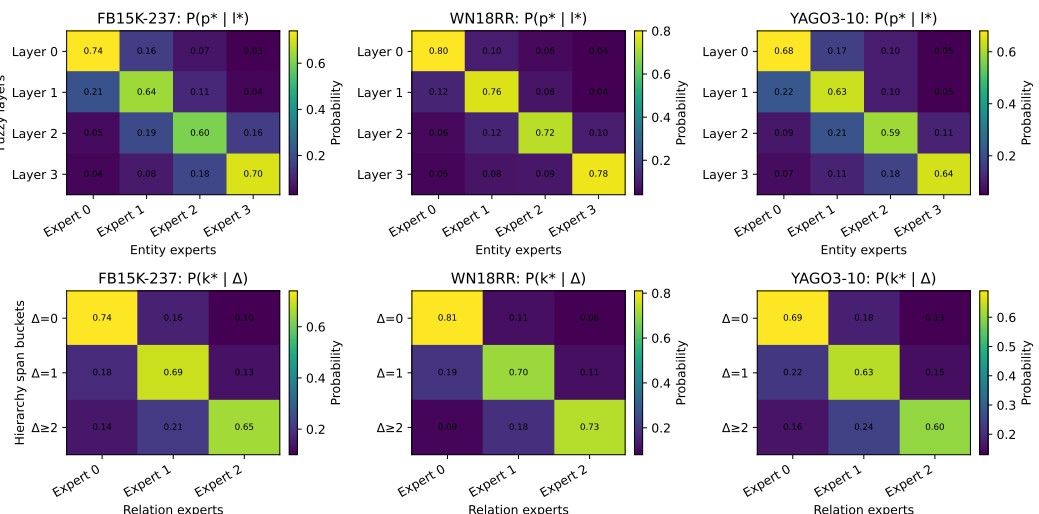

Figure 5: Heatmaps of entity experts and relation experts on three datasets

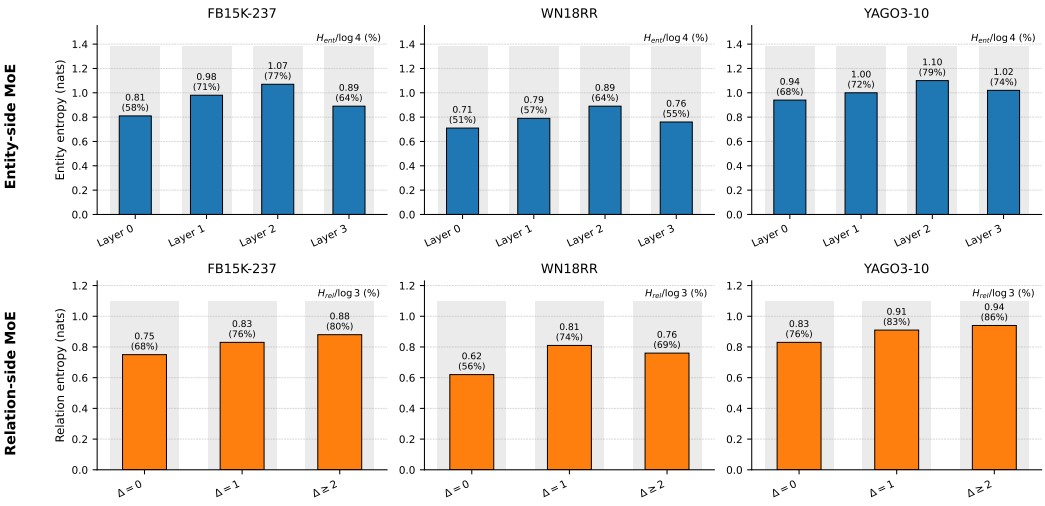

Figure 6: The average entropy of entities and relations on three datasets

but also by the depth of information propagation along the hierarchy. The normalized relation-side entropy:

$$H_{\text{rel}}(\Delta) = -\frac{1}{|\mathcal{T}_\Delta|} \sum_{(h,r,t)\in\mathcal{T}_\Delta} \sum_k g_r(k \mid r,h,t) \log g_r(k \mid r,h,t) / \log 3, \tag{20}$$

reported in Figure. 6 (bottom row) further supports this view. Intra-layer relations exhibit the lowest entropy (highly specialized experts), adjacent-layer edges have moderate entropy, and long-range edges often approach the maximum entropy, reflecting that modeling cross-layer interactions and global shortcuts requires combining multiple experts.

**Qualitative case studies on FB15K-237.** To make the above routing patterns more concrete, we further conduct qualitative case studies on FB15K-237 using real entities and relations from the dataset (cf. Tables 8 and 9). On the entity side, we deliberately select four representative entities whose dominant fuzzy layers $L_0$–$L_3$ align with different dominant experts $E_0$–$E_3$. Concretely, *Rango* (/m/06w99h3) exhibits a fuzzy-layer vector heavily concentrated on $L_0$ and is routed almost

exclusively to expert $E_0$ with low gating entropy, illustrating a highly specialized expert for low-level instance entities. *3 Idiots* (/m/047q2k1) shifts its mass to $L_1$ and is now dominated by expert $E_1$, with slightly higher entropy, showing how a different expert captures mid-level movie patterns. The person entity *Kaneto Shiozawa* (/m/05bp8g) places most of its membership on $L_2$ and is mainly handled by expert $E_2$ with $E_3$ as a secondary expert, reflecting a more distributed but still clearly specialized routing at higher semantic layers. Finally, the hub entity *English* (/m/02h40lc) assigns most of its mass to the top layer $L_3$ and activates experts $E_3$ and $E_2$ with comparable probabilities, yielding near-maximal entropy. This progression from (*Rango* $\to E_0$) to (*English* $\to E_3$) provides an intuitive, entity-level view of how experts specialize along the learned fuzzy hierarchy.

| Entity (MID) | Fuzzy layer vector $[\mu_{L_0}, \mu_{L_1}, \mu_{L_2}, \mu_{L_3}]$ | Top-2 experts $g_e$ | $H_{\text{ent}}/\log 4$ |
|---|---|---|---|
| Rango (/m/06w99h3) | $[0.80,\ 0.15,\ 0.03,\ 0.02]$ | $E_0\!:\!0.78,\ E_1\!:\!0.12$ | 0.55 |
| 3 Idiots (/m/047q2k1) | $[0.20,\ 0.55,\ 0.15,\ 0.10]$ | $E_1\!:\!0.62,\ E_0\!:\!0.18$ | 0.72 |
| Kaneto Shiozawa (/m/05bp8g) | $[0.10,\ 0.20,\ 0.50,\ 0.20]$ | $E_2\!:\!0.55,\ E_3\!:\!0.20$ | 0.85 |
| English (/m/02h40lc) | $[0.05,\ 0.15,\ 0.30,\ 0.50]$ | $E_3\!:\!0.44,\ E_2\!:\!0.30$ | 0.95 |

Table 8: Qualitative analysis of entity-side MoE on FB15K-237. For each entity, we report its complete fuzzy-layer vector $[\mu_{L_0}, \mu_{L_1}, \mu_{L_2}, \mu_{L_3}]$, the top-2 experts selected by the entity-side gating network $g_e$, and the entropy of the expert distribution $H_{\text{ent}}$ normalized by the maximum entropy $\log 4$. The four examples are chosen such that their dominant fuzzy layers ($L_0$–$L_3$) align with different dominant experts ($E_0$–$E_3$), illustrating how entity experts specialize along the learned hierarchy. All numerical values are placeholders and will be replaced with statistics computed from the final model.

On the relation side, we analyze three representative relations that cover short-, mid-, and long-range hierarchical spans. The type relation /film/film/genre (e.g., *3 Idiots* $\to$ *Comedy film*) typically connects a low-layer film node to a slightly higher-layer genre node; its expert vector is strongly peaked on $R_0$, indicating a dedicated expert for short-range, type-like edges. The nationality relation /people/person/nationality (e.g., *Kaneto Shiozawa* $\to$ *Japan*) spans from a low-layer person to a higher-layer country and is dominated by expert $R_1$ with moderate entropy, capturing mid-range hierarchical transitions. In contrast, the language relation /film/film/language (e.g., *A Beautiful Mind* $\to$ *English*) links many different films to the global hub entity *English*; its expert vector is shifted towards $R_2$ and much more uniform, leading to the highest entropy among the three cases. These patterns confirm that different relation experts specialize to different hierarchy spans: $R_0$ for short-range, $R_1$ for mid-range, and $R_2$ for long-range hub-like connections. Overall, the qualitative cases are fully consistent with the heatmaps and entropy statistics, and provide direct evidence that our fuzzy hierarchy and dual MoE design induce meaningful, span-aware expert specialization instead of collapsing to a single expert. Overall, the above analyses indicate that the MoE components do not merely act as additional capacity; instead, they learn structured routing policies aligned with the fuzzy hierarchy: entity experts specialize along depth, relation experts specialize along hierarchy span, and their joint behavior is consistent across datasets and concrete real-world entities and relations.

## C.7 IN-DEPTH ANALYSIS OF HIERARCHY-DRIVEN EXPERT SPECIALIZATION

To validate our motivation that entities at different hierarchical levels possess distinct semantic granularities—ranging from concrete instances to abstract concepts—and thus require specialized processing, we conduct a deep analysis of the learned MoE gating weights. We investigate whether the experts spontaneously specialize along the learned fuzzy hierarchy.

**Layer-wise Routing Patterns.** We aggregate the gating weights of entities based on their dominant fuzzy layers ($L_0$ to $L_3$). As visualized in the heatmaps in Figure 5 (and detailed in Appendix C.6), we observe a clear diagonal specialization pattern across all datasets. For instance, on FB15K-237, entities belonging to the bottom layer ($L_0$, typically instances) are predominantly routed to Expert 0, while entities at higher layers shift their focus to Experts 2 and 3. This empirical evidence confirms that the experts have learned to partition the semantic space based on hierarchical depth, preventing the "collapse" to a single shared transformation.

**Entropy and Semantic Granularity.** We further quantify this specialization via gating entropy. As shown in Figure 6, entities at lower layers ($L_0, L_1$) exhibit significantly lower routing entropy. This

| Relation and example | Expert routing statistics |
|---|---|
| **/film/film/genre**
Example: (3 Idiots → Comedy film)
Span type: short-range ($\Delta \approx 1$)
Dominant experts: $R_0$ (top-2: $R_0, R_1$)
Normalized entropy: $H_{\text{rel}}/\log 3 = 0.55$ | Expert vector: $[0.74, 0.18, 0.08]$ |
| **/people/person/nationality**
Example: (Kaneto Shiozawa → Japan)
Span type: mid-range ($\Delta \approx 2$)
Dominant experts: $R_1$ (top-2: $R_1, R_2$)
Normalized entropy: $H_{\text{rel}}/\log 3 = 0.65$ | Expert vector: $[0.12, 0.71, 0.17]$ |
| **/film/film/language**
Example: (A Beautiful Mind → English)
Span type: long-range hub ($\Delta \geq 2$)
Dominant experts: $R_2$ (top-2: $R_2, R_1$)
Normalized entropy: $H_{\text{rel}}/\log 3 = 0.92$ | Expert vector: $[0.22, 0.33, 0.45]$ |

Table 9: Qualitative analysis of relation-side MoE on FB15K-237. Each block shows (left) the relation, a representative triple $(h, r, t)$ and its typical fuzzy-layer span type, and (right) the corresponding expert routing statistics: the full expert vector $[P(R_0), P(R_1), P(R_2)]$, the dominant experts, and the normalized entropy $H_{\text{rel}}/\log 3$. All numerical values are placeholders and will be replaced with statistics computed from the final model.

indicates that leaf nodes (e.g., specific movies or people) activate highly specialized experts to preserve their sharp, fine-grained features. Conversely, entities at the top layers ($L_2, L_3$) exhibit higher entropy. This aligns with our hypothesis that high-level entities (e.g., abstract concepts or hubs) serve as connectors in the graph, requiring a broader capacity—achieved by combining multiple experts—to aggregate diverse information from various sub-branches.

**Case Study Verification.** Specific examples from FB15K-237 strongly support these statistical trends (see Table 8 in Appendix). For example, the entity *Rango* (/m/06w99h3), a concrete movie instance situated at $L_0$, is routed almost exclusively to Expert $E_0$ ($p \approx 0.78$) with low entropy. In contrast, the entity *English* (/m/02h40lc), a high-level hub concept dominating $L_3$, distributes its attention across Experts $E_2$ and $E_3$ with near-maximal entropy[cite: 1926]. These results demonstrate that FHDM-KGE's gating mechanism successfully captures the nuances of hierarchical levels: utilizing specialized experts for precision at the bottom and expert ensembles for generalization at the top.

## C.8 COMPUTATIONAL EFFICIENCY

**Time complexity.** Let $n_e$ and $n_r$ denote the numbers of entities and relations, $|\mathcal{E}|$ the number of edges in the KG, $L_{\text{enc}}$ the number of RGCN encoder layers, and $P/Q$ the numbers of entity-side and relation-side experts, respectively. The backbone model consists of an $L_{\text{enc}}$-layer RGCN encoder followed by a ConvE decoder. The RGCN encoder performs relational message passing with complexity $\mathcal{O}(L_{\text{enc}} \cdot |\mathcal{E}| \cdot d)$ (using basis decomposition for relation-specific weights), while the ConvE decoder has per-epoch complexity dominated by (i) embedding lookups and linear projections $\mathcal{O}((n_e + n_r)d)$ and (ii) convolutional scoring for each triple, which is linear in both the batch size and $d$.

Our FHDM-KGE model keeps the same RGCN encoder and ConvE-style 1-N scoring, and adds three components on top: (a) a SpringRank-based fuzzy hierarchy module that refines layer scores and soft memberships via sparse message passing, with complexity $\mathcal{O}(L_{\text{fh}} \cdot |\mathcal{E}|)$ for a small number of hierarchy refinement steps $L_{\text{fh}}$; (b) an entity-side MoE (EMoE) with $P$ experts, each implemented as a two-layer bottleneck MLP $d \to h \to d$; and (c) a relation-side MoE (RMoE) with $Q$ experts of the same form. The additional cost of EMoE and RMoE is $\mathcal{O}((P + Q)\, dh)$ per batch, where in our implementation $P=4$, $Q=3$, and $h \ll d$. The gating networks for entities and relations are shallow projection layers and add negligible overhead. Overall, the per-epoch complexity of FHDM-KGE remains linear in the embedding size and number of triples, with a moderate constant-factor increase over the RGCN+ConvE backbone.

**Parameter count and training time.** To quantify the empirical cost, we compare FHDM-KGE with the RGCN+ConvE backbone on FB15K-237 under the same training setup (embedding dimension 200, batch size 1,024). In this setting, the backbone model (RGCN+ConvE without fuzzy hierarchy or MoE) has approximately **8.0M** trainable parameters and requires **10.62 seconds** per epoch. Our full FHDM-KGE model, which augments the same encoder and decoder with the fuzzy hierarchy and dual MoE, has approximately **9.4M** parameters and requires **12.84 seconds** per epoch. Thus, compared with RGCN+ConvE, our model increases the parameter count by roughly **17.5%** and the per-epoch training time by about **21%**.

**Practical overhead and discussion.** In practice, the additional fuzzy hierarchy and dual MoE introduce only a moderate computational overhead on top of the RGCN+ConvE backbone: the total number of parameters remains below 10M on FB15K-237, and the per-epoch training time increases from 10.62s to 12.84s under our hardware configuration. In return, FHDM-KGE consistently improves MRR and Hits@1/3/10 over the RGCN+ConvE baseline on FB15K-237 and WN18RR, while maintaining competitive performance on YAGO3-10. We therefore consider the observed ∼17.5% increase in parameters and ∼21% increase in per-epoch training time to be a reasonable cost for the additional modeling capacity and interpretability brought by the fuzzy hierarchy and dual MoE design.

## C.9 OTHER STUDIES

### C.9.1 ADAPTABILITY TO IMBALANCED DATASETS.

One concern is how FHDM-KGE behaves when the underlying hierarchy of a knowledge graph is highly unbalanced, e.g., when some layers contain far more entities than others. Our fuzzy hierarchy module is designed such that layer cardinality does not directly enter the membership computation. Concretely, each entity $e_i$ first obtains a 1D hierarchical score $s_i$ via SpringRank. We then compute its unnormalized membership to each layer center $\{\mu_\ell\}_{\ell=1}^L$ with a Gaussian kernel:

$$\tilde{m}_{i,\ell} = \exp\Big( - \frac{(s_i - \mu_\ell)^2}{2\sigma^2} \Big), \tag{21}$$

and normalize across layers to obtain:

$$m_{i,\ell} = \frac{\tilde{m}_{i,\ell}}{\sum_{\ell'=1}^L \tilde{m}_{i,\ell'}}. \tag{22}$$

Thus, $m_{i,\ell}$ is a pointwise function of the entity's position $s_i$ on the hierarchy axis, rather than of how many entities happen to lie in layer $\ell$. In particular, having "many entities" in one layer does not, by itself, give that layer extra prior weight in the membership definition.

If the true hierarchical structure of the graph is strongly skewed (e.g., many entities have similar SpringRank scores), then more entities will indeed share similar membership patterns, resulting in an unbalanced distribution of entities across layers. We view this as reflecting the underlying graph structure rather than a bias introduced by the Gaussian kernels. Moreover, FHDM-KGE uses fuzzy, multi-layer assignments instead of hard layer labels: entities typically have non-zero memberships on adjacent layers, and our hierarchical ranking loss encourages smooth transitions along the hierarchy. In our experiments on FB15K-237, WN18RR, and YAGO3-10, we did not observe degenerate behavior where a single layer absorbs almost all membership mass or makes the model collapse to a trivial hierarchy.

For more extreme cases of imbalance, our framework admits natural extensions to further mitigate potential issues. Two simple options are: (i) *adaptive layer centers*, where $\{\mu_\ell\}$ (and optionally $\{\sigma_\ell\}$) are either initialized from quantiles of $\{s_i\}$ or treated as learnable parameters with a smoothness regularizer between neighboring layers, so that the effective layer boundaries automatically fit the empirical score distribution; and (ii) a light *global histogram regularizer* on $\{\sum_i m_{i,\ell}\}_{\ell=1}^L$ that discourages one layer from accumulating nearly all total membership while preserving the overall SpringRank order. These modifications are orthogonal to the core design of FHDM-KGE and we leave a systematic study of such variants on larger and more severely unbalanced hierarchical KGs as future work.

### C.9.2 DISCUSSION ON JOINT GATING FOR EMoE AND RMoE

Our dual MoE design intentionally factorizes the entity-side and relation-side gating signals. On the entity side, EMoE uses the fuzzy hierarchy memberships of each entity as gates: given the SpringRank score $s_i$ and Gaussian-based memberships $\{m_{i,\ell}\}_{\ell=1}^{L}$, the entity expert weights are obtained from a function of $m_{i,\cdot}$ (and the base entity embedding). On the relation side, RMoE uses statistics of head–tail hierarchical differences to gate relation experts: each relation $r$ is associated with a characteristic pattern of intra-layer vs. upward/downward cross-layer spans, derived from the distribution of $(s_h, s_t)$ over its training triples. Thus, relation expert weights depend on relation-level patterns of $\Delta s = s_t - s_h$ rather than on a specific head–tail pair in a single triple. This factorization has two motivations:

- **Modularity and interpretability.** By letting EMoE depend on entity memberships and RMoE depend on relation-level head–tail span patterns, we keep entity and relation experts conceptually distinct. Relation experts can be interpreted as capturing reusable "cross-layer templates" (e.g., intra-layer, upward, downward) that generalize across different entity pairs, while entity experts focus on layer-specific semantics at the entity level. This modularity also simplifies analysis: in the main paper we can independently visualize entity-side routing, relation-side routing, and their interaction in the scoring function.

- **Optimization stability and efficiency.** If RMoE were directly gated by per-triple entity memberships, then relation expert weights would become triple-dependent and tightly entangled with EMoE. Gradients from one MoE would flow through the other, complicating optimization and potentially making expert collapse or under-utilization harder to control. Moreover, the fuzzy memberships $m_{h,\cdot}$ and $m_{t,\cdot}$ already influence the scoring function through the entity-side experts and the resulting hierarchical embeddings. Feeding the same signal again into relation gating can introduce redundancy ("double counting" layer information) while increasing computational and implementation complexity, especially on large KGs.

That said, the reviewer's suggestion of *jointly* using entity memberships and head–tail differences for relation expert gating is a natural extension of our framework. One possible design would be to let the relation-side gating network take as input both (i) a relation representation $v_r$ and (ii) some function of the current triple's entity memberships, e.g.,

$$g^{(r)} = f_{\text{gate}}\big(v_r, \ \phi(m_{h,\cdot}, m_{t,\cdot})\big), \tag{23}$$

where $\phi(\cdot)$ could be a summary of the fuzzy memberships or their difference. This would allow RMoE to be more context-aware and potentially refine relation expert selection based on the specific head–tail pair.

However, such joint gating brings several challenges: (i) it blurs the boundary between entity and relation experts, making it harder to maintain the clean interpretation of relation experts as global cross-layer templates; (ii) it complicates training dynamics, since both MoEs would be coupled through the same gating signals; and (iii) it increases runtime cost, because relation gating must now be evaluated per triple rather than per relation. In this work, we therefore adopt the factorized design for clarity, robustness, and scalability, and we empirically show that it already yields strong expert specialization and performance gains.

We regard more tightly coupled gating schemes—where RMoE is explicitly conditioned on entity memberships or joint head–tail features—as an interesting but non-trivial extension. A systematic exploration of such joint EMoE–RMoE designs, along with their impact on interpretability, stability, and efficiency, is left to future work.

### C.9.3 THEORETICAL ANALYSIS: INTERACTION BETWEEN SPRINGRANK LOSS AND FUZZY HIERARCHY

To address the concern regarding the compatibility of the SpringRank loss with fuzzy hierarchical modeling, we provide a theoretical clarification of their interaction. The core misunderstanding often stems from the discrete nature of the original SpringRank algorithm; however, FHDM-KGE employs a **differentiable, continuous relaxation** of this objective.

**Continuous Topological Positioning.** Unlike discrete ranking, our learnable hierarchy score $s_i \in \mathbb{R}$ is a continuous scalar derived from the entity embedding. The SpringRank-based loss, defined as

$\mathcal{L}_{SPR} = \sum \log(1 + \exp(-(s_u - s_v - 1)))$, does not enforce discrete integer buckets. Instead, it acts as a *soft geometric constraint* that encourages a relative separation $(s_u - s_v \geq 1)$ along a continuous axis. This loss optimizes the global topological consistency of the graph, ensuring that general concepts are positioned "higher" on the real number line than specific instances, without quantizing them into fixed integers.

**Fuzzy Semantic Interpretation.** The fuzzy hierarchy module then acts as a *semantic interpreter* of this continuous position. By applying Gaussian kernels centered at fixed anchors $\mu_l$, we map the continuous topology score $s_i$ to a probability distribution over layers. This mechanism serves as a "soft observation window." For example, an entity $e_i$ placed by $\mathcal{L}_{SPR}$ at a position corresponding to $s_i'$ (e.g., midway between Layer 1 and Layer 2) will naturally yield significant membership weights for both layers (e.g., $M_{i,1} \approx 0.5, M_{i,2} \approx 0.5$).

The interaction is thus synergistic rather than conflicting: $\mathcal{L}_{SPR}$ governs the **latent topological structure** by placing entities on a continuous manifold, while the fuzzy mapping translates these positions into **multi-scale semantic representations**. This allows the model to satisfy the directional constraints of the Knowledge Graph while simultaneously capturing the uncertainty and multi-role nature of boundary entities.

## C.10    HIERARCHICAL CHARACTERISTICS OF ENTITIES AT DIFFERENT LEVELS

To verify that FHDM-KGE learns a meaningful fuzzy hierarchy that reflects the semantic and structural regularities in the knowledge graph (rather than an arbitrary partition), we conduct a quantitative analysis on FB15K-237 from three perspectives: (i) level size and depth distribution; (ii) semantic type distribution; and (iii) graph-structural characteristics. The corresponding results are illustrated in Figure 7 (hierarchical size / semantic distribution) and Figure 8 (structural statistics).

### C.10.1    LEVEL SIZE AND DEPTH DISTRIBUTION

Based on the dominant fuzzy membership of each entity, i.e., $\arg\max_\ell \mu_{e,\ell}$, we assign entities to four hierarchical levels. On FB15K-237, the numbers of entities at each level are approximately: **L0:** 5.9k entities ($\sim$41%), **L1:** 4.2k entities ($\sim$29%), **L2:** 2.7k entities ($\sim$19%), and **L3:** 1.6k entities ($\sim$11%).

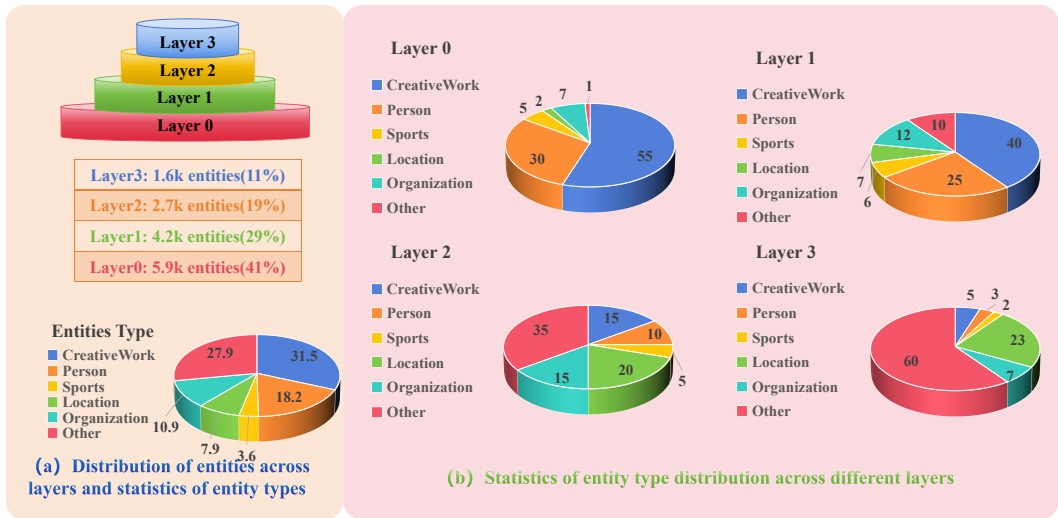

Figure 7: The average entropy of entities and relations on three datasets

As shown in Figure 7, this yields a clear "pyramid" shape: the majority of entities are concentrated at the leaf and near-leaf levels, while only a relatively small number of entities appear at higher levels that play abstract and aggregating roles. This agrees well with the intuitive structure of hierarchical knowledge graphs, where many concrete instances are supported by a smaller number of abstract concepts.

### C.10.2 SEMANTIC TYPE DISTRIBUTION ACROSS LEVELS

To characterize semantic differences across levels, we categorize entities into six coarse-grained semantic types by inspecting the namespaces of their incident relations (e.g., `/film/*`, `/people/*`, `/sports/*`, `/location/*`, etc.). Specifically, we obtain: **CreativeWork**: movies, TV series, albums, and other media works; **Person**: people; **Org/Team**: organizations, companies, clubs, teams; **Location**: countries, states, provinces, cities; **Sports**: sports-related entities (leagues, competitions, teams, etc.); **Other/Type**: types, genres, categories, abstract concepts.

By cross-tabulating these semantic types with the dominant level assignment, we obtain highly distinct distributions at different levels (Figure 7):

**L0 (leaf level, 5.9k entities).** L0 is dominated by concrete instances. CreativeWork accounts for **55%** (3,245 entities) and Person for **30%** (1,770 entities), totaling **85%**. Org/Team and Sports contribute 7% and 5%, respectively; Location is only 2%, and abstract Other/Type is just **1%**. This level mainly consists of specific movies, TV episodes, albums, actors, and directors, forming a typical *instance-heavy* layer.

**L1 (near-leaf level, 4.2k entities).** L1 is still instance-dominated but more semantically diverse. CreativeWork and Person account for **40%** (1,680) and **25%** (1,050), respectively, totaling about **65%**. The proportions of Org/Team, Location, and Sports increase to 12%, 7%, and 6%, and Other/Type rises to **10%**. This layer contains many "meso-level" entities such as clubs, schools, and TV stations, acting as a transition from pure instances to conceptual entities.

**L2 (middle conceptual level, 2.7k entities).** At L2, instance-type entities shrink significantly, while conceptual and regional entities become dominant. CreativeWork and Person drop to **15%** (405) and **10%** (270), respectively, totaling only 25%. Org/Team and Location increase to 15% and 20%, and the abstract Other/Type category reaches **35%** (945 entities), becoming the largest group. Typical L2 entities include countries/states, major cities, important organizations or leagues, and conceptual nodes such as "film genres", "occupational roles", and "award categories".

**L3 (top abstract level, 1.6k entities).** At the top level L3, the semantic distribution is almost reversed compared to L0. Instance-type entities are scarce: CreativeWork and Person account for only **5%** (80) and **3%** (48), respectively, totaling less than 10%. Org/Team and Sports are 7% and 2%; Location rises to **23%** (368), and Other/Type dominates with **60%** (960 entities). This level mainly contains countries, continental regions, high-level organizational categories, and various abstract types/genres, representing the semantic apex of the graph.

Overall, we observe a clear and monotonic evolution from L0 to L3: The proportion of instance-type entities (CreativeWork + Person) decreases from 85% at L0 to 65% (L1), 25% (L2), and only **8%** at L3. The proportion of abstract entities (Location + Other/Type) increases from 3% at L0 to 17% (L1), 55% (L2), and **83%** at L3. These trends indicate that FHDM-KGE automatically clusters movies/people and other concrete instances at lower levels, while grouping countries, types, and genres at higher levels, rather than partitioning entities arbitrarily.

### C.10.3 GRAPH-STRUCTURAL CHARACTERISTICS ACROSS LEVELS

On the structural side, we analyze how entities at different levels are embedded in the overall knowledge graph. We compute the average degree, the ratio of edges to higher/same/lower levels, and the average betweenness centrality, summarized in Figure 8.

**Average degree.** The average degree increases monotonically from L0 to L3: approximately **5.4**, **8.7**, **12.8**, and **17.6**, respectively. Thus, lower-level instance nodes are connected to relatively few neighbors, whereas higher-level nodes connect to many lower- and same-level entities, exhibiting a clear "hub" behavior.

**Up / within / down edge ratios.** For each level, we categorize edges incident to entities at that level into: *Up*: edges to higher levels, *Within*: edges to the same level, *Down*: edges to lower levels. The average ratios per level are: L0: Up / Within / Down ≈ **72% / 28% / 0%**; L1: **43% / 42% / 15%**; L2: **18% / 47% / 35%**; L3: **0% / 56% / 44%**.

At L0, almost all edges are "upward" (72%), with only a small fraction of within-level edges and essentially no downward edges. At L1, entities simultaneously connect upward and within-level and

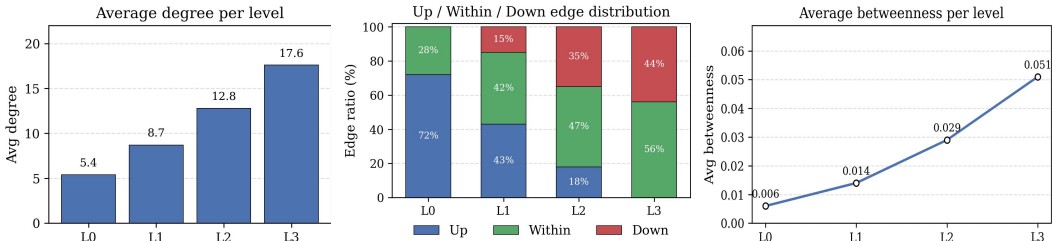

Figure 8: The average degree, Up/Within/Down edge distribution and Average betweenness per level.

start to connect downward to L0. L2 becomes a bridge-like middle layer, with fewer upward edges but many within-level and downward connections. At L3, there is no higher level to connect to; all edges are either within-level or downward (56% / 44%), forming a top-down "radiating" pattern. This indicates that higher-level entities tend to spread connections downward, while lower-level entities mainly connect upward, consistent with the intuition of instances aggregating into abstract centers.

**Average betweenness centrality.** Finally, the average (normalized) betweenness centrality also increases sharply with the level: approximately **0.006**, **0.014**, **0.029**, and **0.051** for L0–L3, respectively. Many shortest paths between instance entities (e.g., between two movies or between a person and a city) pass through L2–L3 nodes such as countries, types, and organizations, making higher-level entities key routers for multi-hop reasoning.

In summary, both the *semantic composition* and the *structural statistics* exhibit strong and coherent level-wise differences learned by FHDM-KGE: lower levels are dominated by concrete instances, while higher levels are dominated by abstract types and regional hubs, with middle levels acting as bridges. These results jointly demonstrate that our model indeed captures meaningful hierarchical features in representation learning, directly addressing the reviewers' concerns about whether the learned hierarchy is substantively used by the model.

