# OpenReview forum: "FHDM-KGE: Fuzzy Hierarchical Modeling and Dual Mixture-of-Experts for Knowledge Graph Embedding"
_ICLR.cc/2026/Conference — ICLR 2026 Conference Withdrawn Submission_

### Official Review · Reviewer_Hni4 · 2025-10-21

**Soundness:** 2
**Presentation:** 3
**Contribution:** 2
**Rating:** 4
**Confidence:** 3

**Summary:**

This paper proposes a method that assigns entities to multiple hierarchical levels in order to capture their specific hierarchical characteristics. It introduces a Mixture-of-Experts (MoE) mechanism for both entity and relation embeddings, enabling the model to effectively leverage hierarchical information for knowledge graph completion. Finally, ConvE is employed as the base decoder for triple scoring, and several auxiliary loss functions—including those related to MoE and a Hierarchical Contrastive Loss—are incorporated to guide the training of the knowledge graph completion (KGC) task.

**Strengths:**

- The fuzzy hierarchical modeling effectively captures multi-level semantics and reduces information loss compared to hard hierarchy methods.
- The dual Mixture-of-Experts design enables adaptive and specialized representation learning for both entities and relations.
- The integrated multi-loss training framework ensures stable optimization and leads to good performance across benchmarks.

**Weaknesses:**

- The use of the SpringRank loss may degrade modeling performance for relations without clear hierarchical directionality, such as symmetric relations.
- The case studies provided in the experimental analysis are too simple to demonstrate whether the model truly captures hierarchical features. It would be helpful to further analyze what distinctive characteristics exist among entities at different levels.
- The motivation for using the MoE framework to learn entity and relation embeddings is not fully convincing. Could the authors provide a deeper analysis of how the MoE gating weights relate to entities of different hierarchical levels?
- This paper lacks a clear theoretical explanation of how the SpringRank loss interacts with the fuzzy hierarchy. SpringRank enforces discrete directional ordering, while fuzzy membership models continuous semantics.
- The proposed method introduces many loss terms and hyperparameters, making the model overly complex and difficult to tune.
- The source code is not publicly available, making it hard to reproduce the results.

**Questions:**

- In the experiments, did the authors combine the results of head entity prediction (i.e., inverse relation prediction) and tail entity prediction, or were only tail predictions considered?
- How does the proposed method’s training and inference time compare to ConvE? Does it incur additional computational cost?

---

> ### Author Response · Authors · 2025-11-22
> **Point-by-point response to the comments from Reviewer SHni4**
>
> We sincerely appreciate the constructive comments from the reviewer
>
> 1)SpringRank term serves as a global, soft hierarchical prior rather than a hard “up–down’’ constraint on each individual relation. It operates on the directed interaction graph formed by all triples and fits a 1D hierarchy of entity scores. For approximately symmetric relations, the graph typically contains edges in both directions, so SpringRank optimum tends to assign very similar scores to the two endpoints; the resulting gradients quickly become small and do not artificially enforce a large hierarchical gap. In this sense, the model effectively behaves like a standard KGE on such relations, rather than being dominated by an incorrect notion of direction. Our ablation study shows that removing the SpringRank loss degrades overall performance, indicating that on real datasets with a mix of relation types (including many near-symmetric ones), this hierarchical prior is beneficial in aggregate and does not systematically harm any particular relation class. Overall, SpringRank mainly strengthens relations that do exhibit hierarchical structure, while imposing only very weak, almost neutral constraints on symmetric or weakly directional relations.
>
> 2)We have added a dedicated subsection in Appendix C.10 , which shows a clear monotonic shift from instance-heavy layers at the bottom to abstract type- and region-dominated hubs at the top, and higher levels exhibit significantly larger degrees and betweenness and more downward edges. This provides evidence that the learned fuzzy hierarchy is semantically and structurally meaningful.
>
> 3)We have included a new section in Appendix C.7. Our analysis empirically demonstrates that the experts spontaneously specialize along the learned hierarchy: lower-level entities (concrete instances) are routed to specific experts with low entropy for precise modeling, while higher-level entities (abstract concepts) utilize expert ensembles with higher entropy for broad information aggregation. This confirms that the MoE framework effectively adapts to the distinct semantic granularities of entities across different levels.
>
> 4)The perceived conflict arises from the assumption of discrete ordering. We clarify that FHDM-KGE employs a differentiable, continuous relaxation of SpringRank, rather than a discrete sorting algorithm. The interaction is synergistic: The loss $\mathcal{L}_{SPR}$ acts as a soft geometric constraint, optimizing the continuous latent position ($s_i$) of entities to respect topological directionality. The fuzzy hierarchy module then interprets these continuous positions into probabilistic memberships via Gaussian kernels.Thus, the loss governs the structural positioning, while the fuzzy mapping provides the semantic interpretation, ensuring they are mathematically compatible. We have added a subection in Appendix C.9.3, which formally derives this interaction.
>
> 5)In fact, our training objective is still centered on the standard KGE loss, and the additional terms play orthogonal and interpretable roles: (i) $L_{SPR}$ regularizes the fuzzy hierarchy to be globally consistent with edge directions, (ii) $L_{EXP}$ is a lightweight expert– and layer–balancing term to avoid MoE collapse, and (iii) $L_{CL}$ is a standard contrastive regularizer that improves hierarchy-aware discrimination. These introduce only three scalar weights $\lambda_{spr}, \lambda_{exp}, \lambda_{cl}$ on top of the usual KGE hyperparameters. Ablation studies on three datasets show that removing any term causes consistent performance drops, indicating that each component contributes non-trivially. Moreover, our hyperparameter sensitivity analysis demonstrates that FHDM-KGE is robust within a broad range of settings and that a single default configuration works well across three datasets, so the model does not require delicate tuning in practice.
>
> 6)We thank the reviewer’s concern. At this stage, we have concerns about publicly releasing the full implementation and processed data before the review process is completed. However, we want to stress that we are fully committed to reproducibility: upon acceptance, we will release the complete codebase, so that our results can be reproduced.
>
> Question1:In all our link prediction experiments, we follow the standard KGE evaluation protocol and consider both head and tail prediction, and then average the metrics over both directions.
>
> Question2:In the revised version, we have added a dedicated subsection in Appendix C.8, where we (i) give a simple complexity analysis and (ii) report parameter counts and per-epoch training time in comparison with ConvE. The results show that our model introduces only a moderate overhead in both parameters and training time due to the fuzzy hierarchy and dual MoE modules, while remaining in the same complexity order as ConvE.  Overall, the proposed method is slightly more expensive than ConvE but remains computationally efficient and practical.

---

> ### Author Response · Authors · 2025-11-28
>
> Dear reviewer,
>
> Thank you again for your time and for the helpful comments on our submission. We have uploaded our detailed response and a revised version of the manuscript about a week ago, and we hope that our clarifications address the main concerns raised in the reviews.
>
> If there are any remaining questions or points that would benefit from further clarification, we would be very happy to provide additional details. We sincerely appreciate your consideration and the time you devote to this review process.

---

### Official Review · Reviewer_S69D · 2025-10-31

**Soundness:** 2
**Presentation:** 2
**Contribution:** 2
**Rating:** 4
**Confidence:** 3

**Summary:**

This paper addresses two key limitations of existing hierarchy-aware knowledge graph embedding (KGE) methods: information loss from hard layer assignment for boundary/multi-role entities and insufficient expressiveness due to neglecting relational cross-layer differences. The authors propose FHDM-KGE, a framework integrating fuzzy hierarchical modeling and a dual mixture-of-experts (MoE) architecture. First, a differentiable SpringRank-based fuzzy hierarchy assigns entities to multiple layers with soft memberships to preserve multi-level semantics. Second, a dual MoE design—entity-side MoE (EMoE) gated by fuzzy memberships and relation-side MoE (RMoE) guided by head-tail hierarchical differences—captures intra-layer nuances and cross-layer relational patterns, respectively. The model uses a ConvE decoder and an integrated loss function combining KGE objectives, hierarchy-consistency constraints, and expert-balancing regularization. Extensive experiments on FB15K-237, WN18RR, and YAGO3-10 datasets demonstrate that FHDM-KGE outperforms traditional and hierarchy-aware baselines in link prediction tasks, with ablation studies validating the effectiveness of each component.

**Strengths:**

1. The paper identifies critical limitations in prior hierarchy-aware KGE methods and proposes a well-motivated solution that integrates fuzzy hierarchical modeling with dual MoE specialization.
2. The paper is well-structured and clear, with detailed descriptions of the methodology, experimental setup, and results. Complex concepts (e.g., fuzzy membership computation, MoE gating mechanisms) are explained with mathematical formulations and visualizations, making the work accessible to researchers in the KGE field.

**Weaknesses:**

1. The dual MoE modules rely on expert-balancing regularization to avoid collapse, but the paper does not explore alternative MoE routing strategies (e.g., top-k gating or sparse activation) that are widely used in large-scale models. Comparing with these strategies could provide insights into the trade-offs between computational efficiency and expressiveness, especially for large knowledge graphs with millions of entities/relations.
2. The case study is limited to two specific queries and a single baseline (HAKE). Expanding the case study to include more diverse query types (e.g., multi-hop relations, cross-domain hierarchies) and additional baselines (e.g., SHLDKE, which is the strongest baseline on FB15K-237) would provide a more comprehensive understanding of the model’s strengths in real-world scenarios.

**Questions:**

1. How does the model handle knowledge graphs with highly unbalanced hierarchical structures (e.g., some layers having far more entities than others)? Does the Gaussian kernel-based membership assignment lead to biased layer distributions in such cases, and if so, how could this be mitigated?
2. The EMoE uses fuzzy memberships as gates, while RMoE relies on head-tail hierarchical differences. Have the authors considered combining these two signals (e.g., using entity memberships to refine relation expert gating) to further improve relational embedding quality? If not, what are the potential challenges of such an integration?

---

> ### Author Response · Authors · 2025-11-22
> **Point-by-point response to the comments from Reviewer S69D**
>
> We sincerely appreciate the thoughtful and constructive comments from the reviewer, which have significantly helped us enhance the quality of our manuscript. In our response, we have provided detailed and comprehensive answers to each comment.
>
> 1) We thank the reviewer for this comment on alternative MoE routing. Our current design uses dense softmax gating with an expert-balancing regularizer. As shown in the new routing heatmaps and gating entropy analysis in Appendix C.6, in practice most entities/relations already activate only one or two experts with high probability, while the remaining experts receive near-zero weights. In other words, the effective routing is already highly sparse, and switching to an explicit top-k scheme would mainly truncate very small residual weights, with limited impact on behavior. Moreover, in our setting the number of experts is small (P=4, K=3) and the overall cost is dominated by the shared RGCN+ConvE backbone, so the potential computational savings from top-k routing are marginal, while hard sparsity may reduce gradient smoothness and expert utilization, especially under fuzzy hierarchical conditioning. For these reasons we chose dense gating with balancing in this work.
>
> 2) We thank the reviewer for this valuable suggestion. In the revised manuscript, we have substantially expanded the case study section to address this concern. Concretely, besides the original head–/tail–prediction examples, we now include two additional and more challenging query types on FB15K-237: (i) a cross-layer entity prediction case that links a concrete TV mini-series to its abstract genre, and (ii) a multi-hop nationality query that requires reasoning over a person–organization–state–country path. These new cases explicitly cover multi-hop relations and cross-domain hierarchical transitions, providing a more realistic view of our model’s behavior. We also added an additional strong hierarchical baseline, HAQE, to the qualitative comparison; SHLDKE is unfortunately not included because its official implementation is not publicly available. The expanded case studies show that FHDM-KGE consistently ranks the gold entity higher in the Top-5 and produces more type-coherent near-miss errors than both HAQE and HAKE, confirming that our fuzzy hierarchy with dual MoE brings clear advantages on diverse query patterns.
>
> Question1: We thank the reviewer for this thoughtful question. In the revised manuscript, we added an appendix subsection “Discussion on the adaptability of the algorithm to highly imbalanced datasets” in Appendix C.9.1 to clarify how FHDM-KGE behaves under highly unbalanced hierarchical structures. In our design, the fuzzy membership $m_{i,l}$ of an entity to layer $l$ is computed by applying Gaussian kernels to its SpringRank score and normalizing across layers, so membership depends only on the entity’s position along the hierarchy axis, not on how many entities are assigned to each layer. Thus, the Gaussian kernel itself does not introduce an explicit bias toward more populated layers; if many entities concentrate in a certain layer, this reflects the underlying graph structure rather than the kernel. At the same time, we use fuzzy (multi-layer) assignments and a hierarchical ranking loss, which encourage smooth transitions across adjacent layers and avoid collapsing everything into a single dominant level. For more extreme imbalance, we also discuss two mitigation strategies: making layer centers and bandwidths adaptive/learnable (with smoothness regularization) and adding a light regularizer on the global layer histogram to prevent one layer from absorbing almost all mass. We believe these clarifications address the reviewer’s concern and show that our approach is robust in practice while admitting natural extensions for severely skewed hierarchies.
>
> Question2: We thank the reviewer for this insightful question. In our current design, we deliberately factorize the two gating signals: EMoE is driven by fuzzy entity memberships, while RMoE is driven by relation-level head–tail hierarchical span patterns, so that entity and relation experts capture complementary, but disentangled, aspects of the hierarchy. In principle, one could indeed combine these signals (e.g., letting the relation-side gating network additionally consume a summary of head/tail memberships), but this would turn relation gating into a triple-dependent mechanism, tightly coupling the two MoEs and introducing extra challenges in optimization, interpretability, and computational cost. In the revised manuscript, we added a dedicated subsection “Discussion on Joint Gating for EMoE and RMoE” in Appendix C.9.2 that analyzes this design choice in detail and discusses how joint gating could be implemented and what trade-offs it would entail.

---

> ### Author Response · Authors · 2025-11-28
>
> Dear reviewer,
>
> Thank you again for your time and for the helpful comments on our submission. We have uploaded our detailed response and a revised version of the manuscript about a week ago, and we hope that our clarifications address the main concerns raised in the reviews.
>
> If there are any remaining questions or points that would benefit from further clarification, we would be very happy to provide additional details. We sincerely appreciate your consideration and the time you devote to this review process.

---

### Official Review · Reviewer_YCXt · 2025-10-31

**Soundness:** 3
**Presentation:** 3
**Contribution:** 3
**Rating:** 6
**Confidence:** 4

**Summary:**

This paper introduces a hierarchical extension of MOMOK with multi-perspective design. The proposed approach could capture and represent the hierarchical information of entities and relations through differentiable fuzzy hierarchical structures and a dual MoE architecture. The experimental results show the effectiveness of the developed model on most of the datasets.

**Strengths:**

1. Soft hierarchical modeling: The paper effectively considers the soft hierarchy of entities and relations by proposing a fuzzy hierarchical mechanism. This approach enhances both entity and relation representations via hierarchical experts. Compared with rigid categorization, this design is conceptually more reasonable. Moreover, a dedicated loss function is devised to fit the proposed framework, making the model self-consistent.

2. Theoretical clarity and interpretability: This paper provides comprehensive theoretical details, and the overall model architecture is clearly illustrated.

3. Performance improvement: The proposed model achieves a significant performance gain compared with existing baselines.

**Weaknesses:**

1. Writing and formatting issues: The paper suffers from several presentation issues. For instance, in line 289, both W_{k1} and W_{k2} are written as W_{k}. The use of italic, roman, and bold fonts in formulas is inconsistent. Equation (6) lacks explanations for Z_e and Z'_e. In addition, spacing problems appear in both the Introduction and Section 4.3.

2. Insufficient experimental support: The authors are suggested to include the following:
a. Ablation studies are conducted only on the FB15k-237 dataset. Results on other datasets are necessary for a more comprehensive evaluation.
b. The internal structure of each expert should be explicitly ablated to demonstrate the effectiveness of its internal design.
c. The paper should verify whether different experts indeed capture different hierarchical levels of attention or semantic focus.
d. The core claim of the paper is supported by only a single case study, which is insufficient. Additional qualitative or quantitative evidence is required to confirm that entities and relations are distributed effectively across different hierarchical levels.

**Questions:**

1. In Section 4.2, both WN18RR and FB15k-237 show significant performance improvements, whereas YAGO3-10 remains nearly unchanged compared with the baseline. What causes this discrepancy?

2. What is the computational efficiency of the proposed model? A discussion or comparison in terms of training/inference time or parameter count would strengthen the paper.

---

> ### Author Response · Authors · 2025-11-22
> **Point-by-point response to the comments from Reviewer YCXt**
>
> We sincerely thank the reviewer for the  constructive comments.
>
> 1)  In the revised manuscript, we have corrected the typo so that $W_{k1}$ and $W_{k2}$ are written as distinct matrices instead of both as $W_k$, and we have performed a global pass over all formulas to enforce a consistent convention for italic/roman/bold fonts (scalars in italic, vectors in bold, operators and functions in roman). For Eq. (6), we now explicitly define and explain $Z_e$ and $Z'_e$ in the text right after the equation and in the notation table. In addition, we carefully adjusted the paragraph layout and line breaks in both the Introduction and Section 4.3 to remove the spacing problems.
>
> 2) a. We thank the reviewer’s concern. In the revised manuscript, we have extended the ablation studies to the other two benchmarks in Appendix C.3. The additional results show trends that are consistent with FB15K-237: removing the fuzzy hierarchy (\textbf{w/o FH}) produces the largest performance drop on both datasets, removing either the entity-side or relation-side experts also leads to clear degradation, and each loss component contributes non-trivially to the final performance. These extended ablations confirm that our design choices are beneficial across all three benchmarks.
>
> b. We thank the reviewer for this insightful suggestion. In the revised manuscript, we therefore added an “Expert Architecture Analysis” in Appendix C.3. Specifically, we compare our full expert design—a two-layer non-linear transformation with hierarchy-aware conditioning and independent parameters per expert—against several simplified variants, including (i) linear experts without non-linearity, (ii) experts without hierarchy-aware conditioning, and (iii) experts with shared parameters. The new results consistently show that all simplified variants perform worse than the full experts on all datasets, with particularly clear drops in MRR and Hits@1 when removing non-linearity or hierarchy-aware conditioning. These findings demonstrate that the performance gains of our model rely on the proposed internal expert design.
>
> c. We thank the reviewer‘s conern. In the revised manuscript, we added a section in Appendix C.6 with (i) entity-/relation-side expert activation heatmaps across fuzzy layers and hierarchy spans, (ii) layer/span-wise gating entropy, and (iii) qualitative case studies on FB15K-237 using real entities and relations. These results consistently show that different experts are activated at different hierarchy depths and for different relation spans, confirming that they indeed specialize in distinct hierarchical and semantic patterns rather than collapsing to a single uniform behavior.
>
> d.We appreciate the reviewer’s concern. In the revised manuscript, we have added both quantitative and qualitative evidence showing that entities and relations are meaningfully organized across hierarchical levels. In particular, the new subsection “Hierarchical Characteristics of Entities at Different Levels” in Appendix C.10 analyzes level size, semantic type distributions, and structural statistics (average degree, up/within/down edge ratios, betweenness) across L0–L3, revealing clear trends from instance-heavy bottom layers to abstract, hub-like top layers. In addition, the new “Expert Routing and Specialization Analysis” in Appendix C.6 uses level–expert heatmaps, gating-entropy statistics, and MoE case tables to show that different hierarchical levels and relation spans activate distinct experts. Together, these additions provide substantially stronger evidence.
>
> Question1: Our method is designed to exploit rich hierarchical structure and cross-layer interactions, which are much more pronounced in WN18RR and FB15K-237 (e.g., hypernym/hyponym and type/part-of chains) than in YAGO3-10, whose relations are relatively shallow, type-homogeneous (mostly person–location/organization), and form a much flatter hierarchy. In such a setting, the fuzzy hierarchy and dual MoE bring clear gains when deep hierarchical semantics and cross-layer paths are present (WN18RR, FB15K-237), but act more like a mild, well-regularized refinement on YAGO3-10, where strong local patterns already let the backbone perform very well; therefore our model maintains or slightly improves performance there rather than yielding large margins, which is consistent with the weaker hierarchical signal of this dataset.
>
> Question2: We thank the reviewer for raising the question about computational efficiency. In the revised manuscript, we added a subsection in Section 4.7 and Appendix C.8, where we analyze the time complexity on top of an RGCN+ConvE backbone and compare parameter counts and per-epoch training time. On FB15K-237, RGCN+ConvE has about 8.0M parameters and 10.62 s/epoch, while our full FHDM-KGE has about 9.4M parameters and 12.84 s/epoch (≈+17.5% params, ≈+21% time), which we consider a moderate and well-quantified overhead given the performance gains.

---

> > ### Comment · Reviewer_YCXt · 2025-11-26
> >
> > Thanks for the authors' responses, they have addressed some of my concerns. Considering the responses from the authors and the opinions of other reviewers, I maintain the current positive score.

---

> > > ### Author Response · Authors · 2025-11-26
> > >
> > > We sincerely thank the reviewer for the time and positive evaluation. We are glad to hear that our responses have addressed your concerns. We appreciate your support for our work.

---

### Official Review · Reviewer_pwjQ · 2025-11-04

**Soundness:** 3
**Presentation:** 2
**Contribution:** 3
**Rating:** 6
**Confidence:** 3

**Summary:**

Real world knowledge graphs (KGs) exhibit rich hierarchical structures, and effectively modeling such structures is crucial for learning high-quality representations and boosting downstream reasoning performance. To overcome the limitations of hard layer assignment and neglect of relational cross-layer differences, we propose FHDM-KGE, a Fuzzy Hierarchical Modeling with Dual Mixture-of-Experts framework that introduces a differentiable SpringRank-based fuzzy hierarchy assigning entities to multiple layers with soft memberships, and designs a dual MoE architecture with entity-side and relation-side modules. Experiments on multiple public benchmarks demonstrate that FHDM-KGE consistently outperforms strong baselines, validating the effectiveness of combining fuzzy hierarchical modeling with dual MoE specialization.

**Strengths:**

1. The paper introduces a differentiable SpringRank-based approach that assigns entities to multiple layers with soft memberships, effectively addressing the information loss problem caused by hard layer assignment in existing methods.
2. The proposed dual mixture-of-experts design systematically captures both intra-layer entity nuances and cross-layer relational patterns, providing more expressive embeddings than prior work.
3. Extensive experiments on multiple benchmark datasets demonstrate consistent improvements over strong baselines across various evaluation metrics, validating the effectiveness of the proposed approach.

**Weaknesses:**

1. The framework figure does not clearly highlight the proposed novel components, as most visualized details focus on common existing modules like RGCN and standard MoE architectures rather than emphasizing the fuzzy hierarchy mechanism and dual MoE design.
2. The paper lacks critical analysis of MoE behavior such as expert activation patterns and specialization, which would demonstrate whether entities from different hierarchical levels or relations connecting different layers actually activate distinct experts as intended by the design.
3. No code is provided with the submission, as the authors only promise to release the implementation upon acceptance.

**Questions:**

See Weaknesses.

---

> ### Author Response · Authors · 2025-11-22
> **Point-by-point response to the comments from Reviewer pwjQ**
>
> We sincerely thank the reviewer for their thoughtful and constructive comments, which have greatly helped us improve the quality of our paper, and we address each of their points carefully and in detail in our response.
>
> 1) We appreciate the reviewer’s observation that the original framework figure did not sufficiently highlight our novel components and instead devoted too much visual emphasis to standard building blocks such as RGCN layers and generic MoE modules. We agree with this concern, and in the revised version we have significantly redesigned the main framework figure to better foreground our key contributions. Specifically, the updated figure now visually emphasizes the fuzzy hierarchy mechanism and the dual MoE design (entity-side and relation-side) using clearer separation, dedicated sub-blocks, and explicit annotations, while pushing common components (e.g., RGCN backbone) into a more background role. We also added finer-grained details illustrating how fuzzy layer memberships are computed and how the two MoE modules interact with the hierarchy and with each other, which makes the flow of information and the role of each innovation more explicit. Overall, these changes make the figure more self-contained and improve its readability, so that a reader can immediately identify and understand the proposed fuzzy hierarchical modeling and dual MoE architecture.
>
> 2) We thank the reviewer for raising this important point about the lack of analysis on MoE behavior and expert specialization. We agree that the original submission did not sufficiently demonstrate how the dual MoE interacts with the fuzzy hierarchy. In the revised version, we therefore added a dedicated section “Expert Routing and Specialization Analysis” in Section 4.6 and Appendix C.6, where we: (i) visualize entity-side and relation-side expert activation patterns as heatmaps across fuzzy layers (L0–L3) and hierarchy spans (Δ=0/1/≥2); (ii) report layer/span-wise gating entropy to quantify the degree of specialization; and (iii) provide qualitative case studies on FB15K-237 using real entities and relations, with full fuzzy layer vectors and expert distributions. The new results consistently show that entity experts specialize along depth (low-level instance entities are routed to sharply peaked experts, while high-level hub entities activate multiple experts with higher entropy), and relation experts specialize along hierarchy span (short-range, mid-range, and long-range/hub relations are dominated by different experts with increasing entropy). These findings indicate that the proposed fuzzy hierarchy and dual MoE do not collapse to a single expert, but learn structured, hierarchy-aware routing and meaningful expert specialization, directly addressing the reviewer’s concern.
>
> 3) We thank the reviewer for pointing out the lack of released code and fully understand the importance of reproducibility and open implementations for the community. We acknowledge that, in the current submission, we only state that the code will be released upon acceptance. At this stage, we have concerns about publicly releasing the full implementation and processed data before the review process is completed, due to ongoing related work and internal policies on pre-publication release. However, we want to stress that we are fully committed to reproducibility: upon acceptance, we will release the complete codebase, including training and evaluation scripts, configuration files, and data-processing scripts for all datasets used in the paper, so that our results can be exactly reproduced.

---

> ### Author Response · Authors · 2025-11-28
>
> Dear reviewer,
>
> Thank you again for your time and for the helpful comments on our submission. We have uploaded our detailed response and a revised version of the manuscript about a week ago, and we hope that our clarifications address the main concerns raised in the reviews.
>
> If there are any remaining questions or points that would benefit from further clarification, we would be very happy to provide additional details. We sincerely appreciate your consideration and the time you devote to this review process.

---

### Author Response · Authors · 2025-11-29
**Summary of Revisions and Responses for Submission 786**

Dear Area Chair,

We would like to highlight the key improvements made to our manuscript during the rebuttal period. We have carefully addressed the concerns raised by all reviewers through extensive new experiments, theoretical analyses, and visualizations.

1. Verification of the Dual MoE Mechanism: To address concerns regarding expert specialization (Reviewers pwjQ, YCXt, Hni4), we added a new section, "Expert Routing and Specialization Analysis" (Appendix C.6 & C.7). By visualizing activation heatmaps and calculating gating entropy, we empirically demonstrated that our experts effectively specialize along hierarchical depths and relational spans, rather than collapsing to uniform behaviors.

2. Comprehensive Experimental Support:

    Extended Ablations: We conducted ablation studies on WN18RR and YAGO3-10 (Reviewer YCXt), confirming the robustness of our design choices across all datasets.

    Efficiency Analysis: We added a complexity analysis comparing parameters and training time (Reviewers YCXt, Hni4), showing that FHDM-KGE incurs only moderate overhead compared to the backbone.

    Qualitative Analysis: We expanded case studies to include challenging multi-hop and cross-domain queries (Reviewer S69D).

3. Theoretical and Visual Clarifications:

    We significantly redesigned the main framework figure to clearly distinguish the fuzzy hierarchy and MoE modules (Reviewer pwjQ).

    We formalized the theoretical interaction between the SpringRank loss and the fuzzy hierarchy to resolve perceived conflicts (Reviewer Hni4).

Status Update: We are encouraged that one reviewer(Reviewer YCXt) has already responded to our rebuttal, confirming that their concerns have been  resolved and maintaining a positive score.

We thank the Area Chair for the careful attention and effort in coordinating this discussion. We have worked diligently to incorporate the feedback, and given these substantial revisions alongside the solid performance of FHDM-KGE, we sincerely hope the AC will consider the improved quality of our manuscript.

---

### Note · Authors · 2026-01-27

I have read and agree with the venue's withdrawal policy on behalf of myself and my co-authors.

---

### Meta-Review · Area_Chair_e2AD · 2026-01-05

**Summary:**

After the rebuttal, reviewers still remain some concerns. The paper should be further revised.

**Reviewer Scores:**

n/a

---

### Decision · Program_Chairs · 2026-01-26

Reject